# Disambiguating sentiment annotation: A mixed methods investigation of annotator experience and impact of instructions on annotator agreement

Laura E. M. Äyräväinen, Joanne Hinds *, Brittany I. Davidson

School of Management, University of Bath, Bath, United Kingdom

* jh945@bath.ac.uk

**Citation:** Äyräväinen LEM, Hinds J, Davidson BI (2025) Disambiguating sentiment annotation: A mixed methods investigation of annotator experience and impact of instructions on annotator agreement. PLoS One 20(12): e0336269. https://doi.org/10.1371/journal.pone.0336269

## Abstract

Human-annotated datasets are central to the development and evaluation of sentiment analysis and other natural language processing systems. However, many existing datasets suffer from low annotator agreement and errors, raising concerns about the quality of data used to train and evaluate computational systems. Improving annotation reliability demands close examination of how datasets are created and how annotators interpret and approach the task. To this end, we create AmbiSent, a new sentiment dataset designed to capture cases of interpretive complexity that commonly challenge both annotators and computational models. Using a mixed-methods approach, we investigate how annotation instructions influence annotator experience and both inter-annotator agreement (Krippendorff's alpha) and intra-annotator agreement (percent agreement). Two groups of 53 crowdworkers annotated 252 sentences under either detailed or minimal instructions, allowing comparison of inter- and intra-annotator agreement, using a permutation test and an independent samples t-test, respectively. Contrary to our hypotheses, our findings reveal that detailed instructions alone do not ensure more consistent annotations – either across or within individuals. A reflexive thematic analysis of open-ended survey responses further contextualised these findings, offering insights into the annotators' cognitive effort involved and the practical challenges faced. Drawing from both quantitative and qualitative findings, the effectiveness of instructions appears contingent on participants' level of task engagement and the extent to which the instructions align with intuitive annotation strategies. Despite the detailed guidance, participants often resorted to reductive annotation approaches. However, we also observed sentence types where detailed instructions may improve annotator agreement, (e.g., sentences with perspective-dependent sentiment, and rhetorical questions). Together, these results inform recommendations for enhancing task engagement and instruction adherence, offering practical insights for future dataset development. Finally, to support diverse

**Data availability statement:** All relevant data are within the manuscript, its Supporting Information files and the following links on the Open Science Framework. This includes: - study resource: https://osf.io/a9vyb/ - detailed description and stimulus preparation: https://osf.io/edg6r/files/osfstorage - other supplementary materials: https://osf.io/r84qj/files/osfstorage - study protocol: https://osf.io/af2h9/files/osfstorage - AmbiSent dataset: https://osf.io/x687m/files/osfstorage.

**Funding:** This work has been funded by the UK Government awarded to BID & JH. The funders had no role in study design, data collection and analysis, decision to publish, or preparation of the manuscript.

**Competing interests:** The authors have declared that no competing interests exist.

use cases, we release three versions of the AmbiSent dataset, each accompanied by detailed annotator information and label distributions to better accommodate different user needs.

## Introduction

Sentiment Analysis is a computational method for classifying subjective information, such as expressions of emotion, opinion and attitudes, in text and other content. Research interest in sentiment analysis is increasing rapidly [1–3], and the potential for real-world applications is met with enthusiasm across sectors, such as finance [4], education [5], and healthcare [6]. Given the widespread application of sentiment analysis in everyday technologies and its increasing impact on decision making, including high-stakes scenarios such as stock market prediction and healthcare applications, it is crucial to ensure that the outcomes from different sentiment analysis systems are valid and trustworthy. A sentiment analysis system is typically assessed by comparing its output against a 'ground truth' dataset, which contains text samples annotated with sentiment labels by human annotators. These datasets (e.g., SemEval 2016 – Task 4 [7], SentiStrength Twitter [8]) not only serve as benchmarks for evaluating system performance, but also as training data that help the system learn to classify new, unseen text. Thus, human judgement forms the foundational standard for sentiment classification in computational systems.

However, human annotators do not always agree on the appropriate sentiment label for a given text sample. In fact, discrepancies in sentiment labels are prevalent [9], and several datasets annotated for sentiment contain annotation errors [10,11]. Apart from the inherently subjective nature of sentiment interpretation, discrepancies in annotations can arise from several other sources, such as fatigue and lapses of attention, different interpretations of the task itself [12], inconsistent annotation strategies adopted, or the order in which items are presented during the task [13]. Apart from disagreements between annotators, there are also intra-annotator discrepancies, that is, the same annotator may assign a different label to the same item, when annotating on two separate occasions [14,15].

As sentiment analysis system development and evaluation rely on annotated datasets, yet, these datasets often suffer from annotation discrepancies, it is essential to understand how these datasets are produced. In this work, we investigate annotator experience using a mixed-methods approach, aiming to discover how the comprehensiveness of instructions may influence annotator agreement and experience, and which aspects of the task and which item types are the most challenging for annotators. Specifically, we focus on interpretative ambiguity in sentiment annotation, which we define as the potential for a sentence to be understood in multiple ways, particularly regarding its emotional or communicative intent. This kind of ambiguity is not lexical or syntactic, but pragmatic – it emerges from the openness of the utterance to multiple context-dependent interpretations. This concept is grounded in established theoretical traditions, such as Gricean pragmatics [16], where flouting or violating conversational maxims can produce ambiguity that listeners resolve differently

depending on inferred speaker intent, and relevance theory [17], according to which listeners infer meaning by selecting interpretations that optimize relevance and cognitive effort. In sentiment annotation tasks, annotators may be aware of and actively choose between competing, plausible interpretations of a sentence, but might not do this consistently for similar types of sentences. We therefore explore whether annotation instructions facilitate a more consistent approach to handling interpretive ambiguity.

Insights from these investigations should help with future construction of datasets, highlighting aspects of the annotation process that require more attention from the research community. Additionally, we introduce AmbiSent, a new sentiment dataset specifically designed to capture interpretive ambiguity in sentiment annotation. The dataset includes soft labels that reflect annotation variability, and detailed annotator demographics, making it a valuable resource for research in sentiment analysis and dataset construction. Our study and the AmbiSent dataset highlight often-overlooked assumptions about subjective annotation work, particularly regarding how annotation tasks are interpreted, what constitutes reliable agreement, and how the 'correct' label is defined. Left unchallenged, these assumptions can compromise efforts to build more robust and inclusive datasets, which are foundational to the development of trustworthy and effective AI systems. As demand for high-quality training data continues to grow in data-intensive AI, it becomes increasingly important to examine how such datasets are constructed.

## Previous investigations of annotator behaviour and experience

Despite the essential role annotators play in shaping datasets, relatively little attention has been paid to how they engage with annotation tasks in practice. Understanding annotator behaviour and experience is crucial for improving annotation quality, such as annotator agreement, error rates, and fidelity to the intended ground truth. This would, by extension, enhance the reliability of systems trained on such data. Optimising the annotation process requires knowing how annotators might perform the task and what aspects of the task they find the most challenging. Some studies have investigated this in different annotation tasks, such as semantic frame annotation [18], work order data annotation [19], and word sense annotation [20]. These studies employed different approaches for tracking annotator behaviour and experience, such as time spent on the task, self-assessed confidence in annotation for each item, and inter-rater agreement measures.

A small number of studies have also investigated annotation as a cognitive process, via eye-tracking during annotation of predicate-argument relations [21], or sentiment annotation [22,23]. These eye-tracking measures were either used to gain insights into the annotation process, (e.g., cognitive processes involved in annotation and their order during annotation), or leveraged in training a classifier using gaze patterns (e.g., fixations and saccades) as additional signals to improve subjectivity extraction and sentiment prediction.

For sentiment annotation in particular, a small number of qualitative studies have explored annotator experiences [24–26], in which free-text feedback on the sentiment annotation task has been used to identify recurring challenges. These studies report that annotators frequently found the task difficult and/or boring, that lack of context made the task particularly challenging, and that the annotators were sometimes uncertain about what should be annotated (such as whether to focus on the sentiment directed towards a particular subject, the speaker's emotional state, or another aspect of the text).

Taken together, despite the central role that human annotation plays in training and evaluating sentiment analysis systems, relatively little work has directly examined the annotators themselves with regards to their instructions, experiences, and interpretation strategies. This lack of attention limits our understanding of how annotation quality is shaped in practice, which impacts the reliability of sentiment models and the validity of their real-world application. Without insights into how annotators interpret and carry out the task, key assumptions about label consistency, task difficulty, and annotator intent may go unexamined – ultimately weakening the foundations of sentiment analysis systems. In the following section, we outline key aspects of the annotation process that may influence data quality and explain how our study addresses these challenges.

### Rationale for the current study

Although human-annotated data is essential to sentiment analysis, the choices made during the annotation process are rarely examined in detail. These include, for example, decisions about whose sentiment to prioritise when conflicting sentiments are expressed, or whether to consider implied sentiment. Understanding such choices is critical for improving both dataset quality and system performance. Below, we outline key aspects of the annotation process, illustrate how these are addressed in prior dataset projects, consider how overlooking them may compromise annotation quality in the produced datasets, and finally describe how we address each aspect in the current study.

### Instructions

Given widely discussed ambiguities related to judging sentiment, clear and comprehensive task instructions are critical for ensuring a uniform understanding of the task itself, thereby reducing annotation discrepancies arising from different task interpretations. The importance of instructions has been neglected in previous sentiment annotation projects [7,27], where annotators have been provided with minimal instructions that lack key information for performing the task, such as specifying what is meant by 'sentiment' and examples for each sentiment label annotators are expected to choose from. However, evidence from different annotation tasks highlights the importance of instructions for annotator agreement [28,29]. For instance, Bayerl and Paul [28] demonstrated with their meta-analysis that inclusion of instructions and higher intensity of annotator training improved annotator agreement in different linguistic annotation tasks. This evidence leads us to make the following assumption: in the absence of specific guidance, annotators are likely to apply their own, potentially inconsistent, criteria – leading to greater annotation variability both within and between individuals. While we do not suggest that detailed instructions eliminate all ambiguity or subjectivity in sentiment annotation, they are expected to reduce interpretative freedom and inconsistency regarding certain aspects of the annotation task, such as whose sentiment to prioritise or how to approach conflicting sentiments.

In our study, we therefore investigate whether the types of minimal instructions found in the sentiment analysis literature result in lower annotation agreement than more detailed instructions. We also explore whether the instruction condition (detailed or minimal) influences how diverse reasons annotators give for their annotation choices.

### Item characteristics

Sentiment datasets often consist of user-generated text samples, such as those scraped from social media (e.g., SemEval 2016 – Task 4 [7], SentiStrength Twitter [8], Stanford Twitter Sentiment [30]). These text samples are often not chosen based on the type of linguistic structures or particular linguistic phenomena they represent, although they may be classified according to these characteristics during data annotation [10]. Datasets scraped from social media data and other similar sources have the benefit of capturing naturally occurring language use. However, the use and distribution of datasets derived from these sources requires navigating complex ethical and legal considerations [31,32]. For example, researchers must mitigate the risk of including material scraped without consent (and have the ability to remove individuals from the datasets at request), alongside managing uncertainty over whether data can be shared with other researchers due to frequent changes in the terms of use for different social media platforms [31].

To avoid these ethical and legal concerns, and to ensure our dataset exhibits linguistic characteristics aligned with the goals of this study, we use purpose-built data generated by crowdworkers. Hence, we construct a dataset of ambiguous cases in sentiment detection, to better understand the challenges faced by human annotators. The items in this dataset are sentences in which sentiment can plausibly be interpreted in different ways. For instance, a sentence with contradicting sentiments from different opinion holders, such as 'I love apples, but Tom dislikes them', could plausibly be interpreted as either 'mixed' (considering both opinion holder's sentiments) or as 'positive' (considering only the stronger sentiment or by prioritising the speaker's sentiment). These types of items are generated by crowdworkers for this specific purpose,

thus ensuring both consent for using the text samples and that the type of text generated is suitable. The instructions for generating the sentences were detailed, with a general requirement for natural-sounding and varied sentences, along with specific requirements for each sentence type, such as 'Write a sentence that expresses a negative sentiment via sarcasm, that is, the literal interpretation might sound positive, but the meaning of the sentence is actually negative.'. Three example sentences (e.g., 'Great phone, the battery died after just two hours') were provided for each sentence type to ensure the crowdworkers understood the desired sentence characteristics.

## Annotator characteristics

In popular datasets, information provided about the annotators is often limited to the type of annotators (e.g., domain experts, crowdworkers) and their native language [7,27]. Such limited information fails to account for ways in which different characteristics and experiences of the annotators influence their interpretation of sentiment. For instance, the mean sentiment scores assigned to a dataset differ by age, gender, and ethnicity of the annotators [33], and the level of agreement between annotators also differs by demographic groups [9,33]. More generally, several studies have demonstrated the impact of annotator characteristics on other interpretation-reliant tasks, such as hate speech or toxicity detection [34,35].

In this work, we collect comprehensive demographic details of the annotators (e.g., age, gender, socio-economic status, variant of English spoken natively), which is presented as group-level summaries in our dataset documentation. This is to provide sufficient context for our dataset, drawing from Bender and Friedman's [36] recommendations.

## Handling discrepancies

Handling disagreements in annotation projects is often based on majority voting or aggregated sentiment scores, in line with the assumption that human consensus is the most appropriate ground truth [37,38]. The released datasets typically contain only the final labels, without information about the level of agreement for each item. This approach is problematic, as it leads to qualitatively different items with the same label, such as unanimously labelled items and those with a much wider range of labels, reduced to the majority or aggregated label [39]. Another common approach is to remove items with low inter-annotator agreement, thereby retaining only those with unanimous or near-unanimous labels. However, excluding items that are difficult for humans to classify also removes the kinds of ambiguity and subjectivity that occur in real-world language, and that sentiment analysis systems should therefore learn to classify. As such, sentiment analysis systems trained and evaluated on such filtered datasets may appear more accurate than they truly are, leading to overly optimistic assessments of their performance and generalisability.

Furthermore, identifying expressions of sentiment in text is reliant on interpretation, and as such, some items have several, plausible interpretations. How an item is interpreted depends on several factors, such as the perspective taken. For instance, the sentence 'England won!' may express positive sentiment if the writer supported the English team, or a negative sentiment if the writer supported the opposing team. In the absence of additional context, both interpretations could be valid. As outlined above, annotators with different demographic characteristics also interpret sentiment differently [9,33]. From an ethical perspective, it is problematic to treat only certain interpretations as 'correct', particularly if alternative interpretations that are potentially held by particular demographic or identity groups are excluded. Sentiment analysis systems trained and/or tested on such datasets can only learn these interpretations, thus leading to systems favouring particular perspectives, while marginalising others.

These issues challenge the assumption that every item must be assigned a single 'correct' label (i.e., a 'hard label'). An alternative is to use 'soft labels', which represent each item as a probability distribution over possible labels, thus capturing annotation uncertainty rather than a single ground truth. Soft labels not only reflect the interpretive nature of sentiment in text more accurately, they can also be beneficial in system development. Rather than collapsing differing judgments into a single ground truth, soft labels preserve the distribution of annotator responses, enabling models to learn

from uncertainty instead of treating it as noise [40,41]. This can lead to improved generalisation to unseen data [41]. For instance, training on soft labels tends to improve model robustness to adversarial attacks (i.e., subtle input manipulations that mislead models into incorrect predictions) [40,42]. These benefits arise because soft labels encourage models to maintain calibrated confidence levels and avoid overfitting to noisy or ambiguous examples [43,44]. They also help mitigate the effects of label noise, as soft targets reduce the influence of outlier annotations that might otherwise skew model training [45]. Moreover, by encoding degrees of annotator disagreement, soft labels promote a more epistemically transparent and psychologically realistic approach to classification, particularly in tasks involving subjectivity or ambiguity [46]. This not only improves model interpretability but aligns more closely with ethical practices in artificial intelligence (AI), where acknowledging uncertainty is often preferable to enforcing false certainty [47].

In this work, we investigate how consistently annotations are produced across different annotators (inter-annotator agreement). Our resulting dataset contains both soft and hard labels for each item, which provide more information about certainty of the labels, and allows more flexible use of the dataset for different purposes.

## The contribution of the AmbiSent dataset

As outlined above, the AmbiSent dataset was developed to address several limitations of existing sentiment datasets: (1) Interpretive ambiguity is typically overlooked, as datasets developers provide minimal instructions, leaving annotators to resolve this ambiguity on their own; (2) Ambiguous items are not systematically included in these datasets, when content is typically scraped from social media and other online content; (3) Annotation agreement is not reflected in the labels, when low-agreement cases are excluded or reduced to a single aggregate label; (4) Important annotator characteristics are rarely reported, despite evidence that demographic differences affect interpretation; and (5) Ethical and legal concerns are common in datasets scraped from social media, where content is often reused without consent and cannot always be redistributed to other researchers. AmbiSent mitigates these issues through purpose-built, consented data collection, controlled inclusion of ambiguous items, detailed task instructions, rich annotator demographic information, and soft labels reflecting annotator variability. These characteristics make Ambisent a unique resource for studying sentiment analysis.

## Research aims of the current study

In summary, the main contributions of our work are two-fold: (1) producing a sentiment dataset and (2) investigating the annotation task as a cognitive process.

First, we construct AmbiSent, a sentiment dataset consisting of ambiguous sentences (i.e., sentences whose sentiment can plausibly be interpreted in different ways). These sentences are annotated by two separate groups of crowdworkers, resulting in two versions of the dataset: one annotated by crowdworkers who received detailed instructions for handling the ambiguity in the sentences, and one annotated by crowdworkers who relied on minimal instructions. Both datasets are also accompanied with demographic information about the annotators, as well as both hard and soft labels for each sentence.

Second, we investigate the role of instructions (detailed or minimal instructions) on annotator agreement, and explore annotator experience during the annotation task. Our hypotheses regarding the role of instructions are as follows:

**H1.** Detailed annotation instructions result in a higher inter-annotator agreement than minimal annotation instructions.

**H2.** Detailed annotation instructions result in a higher intra-annotator agreement than minimal annotation instructions.

Annotator experiences in the annotation task will be examined through exploratory analyses of questionnaire responses and annotator agreement metrics, addressing the following research questions:

**RQ1.** What aspects of sentiment annotation are the most challenging for the annotators, across all annotators and within each instruction group?

**RQ2.** Which sentence types result in the lowest inter-annotator agreement across all annotators and within each instruction group?

**RQ3.** Does the diversity of reasons for annotation choices differ between the instruction groups?

## Method

### Resource availability

The resources associated with this study are available on Open Science Framework (OSF, here). These include a detailed description of the stimulus preparation (here) and all other supplementary materials (here). The study was pre-registered on the 18th of December 2024, and the original protocol document can be found here. Following a pilot study, the protocol was updated on the 17th of March 2025 (available here). Our dataset, AmbiSent, is available here.

### Ethical statement

All parts of the study involving human participants have received ethical approval by the University of Bath Data & Digital Science Research Ethics Committee (reference: 6756–7997 for stimulus preparation, and 6994–8280 for the main annotation study). All participants provided a written consent electronically after receiving information about the study, and before commencing the task. Participants were informed of their right to withdraw from the study at any time, and their right to withdraw their data within seven days after completing the study. A final confirmation of consent was obtained from all participants at the end of the study, after which they were debriefed about the objectives of the experiment and provided with contact details of the research team for further information.

### Procedure

To meet our research aims, we constructed the AmbiSent dataset comprising 252 items (sentences) produced by crowdworkers. These items represent several sentence types that are considered ambiguous or challenging to annotate for sentiment [12]. To ensure that these crowdworker-generated sentences fit the set criteria (e.g., positive sentiment expression via sarcasm, expression of a negative sentiment via a rhetorical question), they were independently evaluated for validity by three native English speakers and included in the final dataset based on majority vote. The included sentences in the final dataset are thus accepted by at least three native English speakers: the sentence author and at least two independent raters who confirmed that each sentence expresses the given sentiment in accordance with the set criteria (for detailed description of the stimulus preparation, see OSF). The sentiment label of each sentence corresponds to the sentiment it was judged to express appropriately (by at least three independent native English speakers), and we refer to these labels as the initial ground truth. These ground truth labels were used as an indicator of instruction comprehension and adherence in the annotation task described below.

The next phase of the study served two purposes: to produce soft labels for the dataset, and to investigate the annotator experience and the role of instructions on annotator agreement. To this end, the dataset was next annotated by a new, large sample of participants. These participants were recruited for a two-session study via Prolific. The succession of tasks in these two sessions is shown in Fig 1, and described in detail in the following sections (session 1 and session 2). The experiment employed a between-subjects design, in which participants were randomly assigned to one of two instruction conditions: detailed instructions or minimal instructions. The remainder of the procedure was identical across conditions. This design was chosen over within-subjects approach to avoid potential carryover and order effects that could

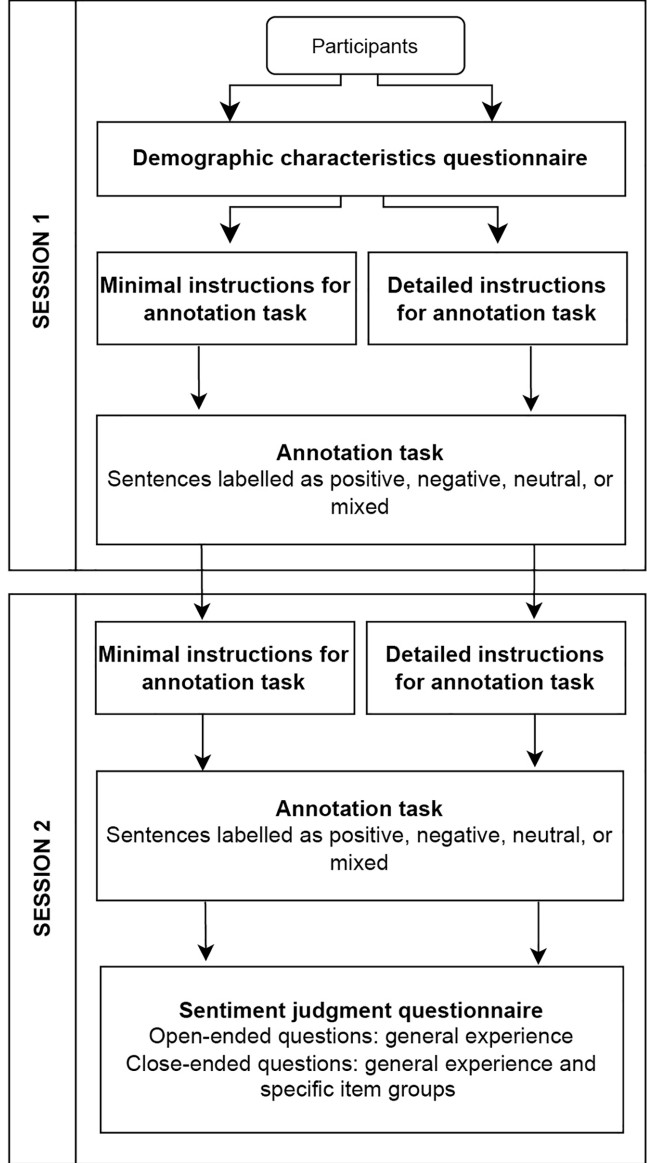

**Fig 1. Task sequence across two sessions of the study.**

arise from exposing participants to both types of instructions. This also allowed for capturing instruction-specific annotator experiences.

**Session 1**

All participants filled in a demographic characteristics questionnaire, after which they read the instructions and annotated seven practice items before commencing the annotation task. The participants could consult the instructions throughout the task. The items in the annotation task were grouped into four blocks, with both item order and block sequence randomized for each participant.

## Session 2

Seven days after session 1, the participants completed session 2. Each participant re-read the same task instructions they were provided with in session 1 (either detailed or minimal instructions). Subsequently, the participants again completed the annotation task, which consisted of a randomly chosen sub-set of items presented in session 1. The same subset of 72 items was presented to all participants, in a randomised order. Following the annotation task, the participants filled in a sentiment judgment questionnaire, consisting of three parts. The three parts of the questionnaire were presented in a fixed order (e.g., Part 1, Part 2, Part 3), as were the open-ended questions in Part 1, whereas multiple-choice questions in Parts 2 and 3 were presented in a randomised order. For open-ended questions, we required a minimum of 30-word responses (150 characters), to set clear expectations for respondents about what constituted an adequate response, and to increase the chances of more in-depth responses to the open-ended questions.

## Data collection and compensation

Data collection took place between 19th of March and 3rd of April 2025. The participants were paid £11.44/hour (in accordance with the UK minimum wage at the time of data collection) to complete the first session in 90 minutes and the second session in 70 minutes. Successfully completing the tasks faster than this resulted in a higher hourly rate (as compensation on Prolific is based on a fixed amount per task regardless of the actual time spent). Any participant who took longer than this and participants who were excluded from the study (see Exclusion of participants) received compensation at the same hourly rate, calculated pro rata based on the precise duration of their task involvement. Thus, we ensured a minimum compensation of £11.44/hour for all participants.

## Materials

### Screening questions

Convenience sampling was used to recruit participants on Prolific. Only those who were at least 18 years old, native English speakers, and without language related disorders (e.g., dyslexia) were invited to participate. Quota sampling was then applied to ensure even numbers of female and male participants.

### Demographic characteristics questionnaire

The purpose of this questionnaire was to provide necessary context for the dataset, so that those using it can evaluate whether it is appropriate for their purposes, e.g., whether the background of the annotators matches the intended use of the dataset. As our sample was international, including speakers of any variant of English, the questionnaire was designed to remain as general as possible for questions that are typically nation-specific (such as terminology used to describe levels of education).

The items included in the questionnaire were motivated by guidelines for dataset documentation [36]. These include factors known to affect language use and proficiency, such as age [48,49] and level of education [50].

In addition to age and native language, we asked which variant of English the respondents spoke natively [36]. This was assessed via a close-ended question, with response options corresponding to national-level variants (e.g., American English, British English), rather than more specific, regional distinctions. The response options also included 'Other' with a free-text response option.

Two questions were presented regarding gender, where respondents were asked to indicate their gender identity and their sex assigned at birth. This was to avoid non-response bias and mis-classification of gender minorities [51] and because gender identity, just like other identity information, should be relevant for language use. More broadly, recent work demonstrates current issues and why more attention should be paid to gender minorities in natural language processing research [52].

Level of education was measured using broad categories from the International Standard Classification of Education [53] as response options. When applicable, the corresponding national level in the United States and/or in the United Kingdom were included to help respondents not familiar with the classification terminology.

Race or ethnic group was tracked via custom-built response options, including 'Other' with a free-text response option. In the absence of widely recognised international standards for race and ethnic groups, we combined the broad categories from standards for collecting and presenting race/ethnicity data across federal agencies in United States [54] and the list of ethnic groups based on The Office for National Statistics 2021 Census of England and Wales [55].

Socio-economic status (SES) was measured as a subjective SES with the Scale of Subjective Status [56,57]. This measure was chosen as it allows cross-national comparisons (compared to annual household income, for instance) and avoids politically charged terms such as 'middle class' and 'working class'. The question was modified to have respondents assess their status in their country of residence.

## Instructions

Two sets of instructions were constructed: the detailed and the minimal instructions. Each participant was presented with one of these instructions, in both testing sessions.

Minimal instructions consisted of a brief task description, without specifying what is meant by different sentiment labels or providing examples of each label (positive, negative, mixed, neutral). A total of two examples were provided (for positive and mixed sentiment). These instructions were inspired by those used for the construction of popular sentiment datasets [7,27], which we imitated regarding the level of detail provided and the number of examples for sentiment labels given. However, our instructions were for sentence level annotation, and as such do not contain instructions regarding targets of sentiments, unlike our inspiration. The minimal instructions were as follows:

> You will see different sentences on the screen. Your task is to decide whether each sentence has a positive, negative, neutral or mixed sentiment. For example, "I like warm summer evenings" has a positive sentiment.

> "Mixed" label refers to a sentence that has a mixed sentiment. For example, the sentence "I liked the food, but the waiter was rude" has a mixed sentiment.

> Try to be objective with your judgement and feel free to take a break whenever you feel tired or bored.

Detailed instructions consisted of defining what is meant by sentiment, and three examples for each of the four sentiment labels. Additionally, instructions were provided for nine ambiguous cases, such as those where a sentence contains contradicting sentiment towards two different targets (e.g., 'I liked the food, but the waiter was rude.') or where a sentence contains a contradicting explicit sentiment expression and an implied sentiment (e.g., 'She was relieved that John got fired.'). Instructions for assessing the strength of sentiment, when necessary, were also provided. These instructions were inspired by common difficulties annotators have reported in previous studies [25,26] and Mohammad's [12] guidelines for constructing clear sentiment annotation instructions. For the purposes of annotation, we defined sentiment broadly as an opinion, attitude, appraisal, feeling, or emotion expressed in text – intended to capture a wide range of subjective and evaluative expressions. This avoided requiring annotators to differentiate between closely related constructs such as emotion, affect, and feeling. While these terms are often used interchangeably in sentiment analysis literature [1], more nuanced distinctions are drawn in psychology, linguistics, and related fields [58–60]. However, requiring annotators to apply such fine-grained distinctions was considered impractical, given the substantial cognitive demands already imposed by the detailed annotation instructions.

Before describing the detailed instructions, we outline some of the ambiguities associated with the sentiment annotation task, specifically for sentence-level annotations. Sentiment may be inferred from text based on different cues, such as

explicit expression of the speaker's emotional state ('I'm disappointed' = negative), appraisal dependent descriptions ('the project was approved' = positive), descriptions of opinions held by someone else than the speaker ('my girlfriend loved it' = positive), sarcasm ('I love being criticised first thing in the morning!' = negative), or rhetorical questions ('Why must we argue all the time?' = negative). Importantly, a single sentence may also include contradicting sentiment cues, such as 'I'm disappointed that the project was approved' or 'my girlfriend loved it but it wasn't my cup of tea'. Furthermore, there are other ambiguous cases, such as perspective-dependent sentiment (e.g., 'England beat France' = positive for supporters of England, negative for supporters of France), and different sentiment towards different targets (e.g., 'The food was good, but the waiter was rude' = positive for food, negative for service).

Clear instructions are required for how to annotate these ambiguous cases. To provide such instructions, decisions are needed regarding how much interpretation is required from the annotators, which sentiment cues to prioritise and whose sentiment or perspective is prioritised. As stated by Mohammad [12], the detailed guidelines for annotation should reflect the intended use, and as such, our definition of what should be annotated is simply one potential solution, which may not be suitable for the requirements in other annotation projects. The main goal for our annotation project was to produce a dataset containing sophisticated human interpretation of sentiment expressed in text, thus reflecting the 'state-of-the-art' of human abilities in this task (as this is, ideally, what sentiment analysis systems should classify from text). We thus aim to capture annotators' pragmatic understanding of texts, which require interpretation of the intended meaning of the speaker, including choices regarding contradicting sentiment expressions. This approach is similar to Mohammad's [12] guidelines, with the overall aim to capture the dominant sentiment of texts. However, unlike Mohammad [12], we specify that the sentiment of both the speaker and another opinion holder should be considered equally important when choosing the sentiment label. This choice is informed by the general principle of intentionality in speech act theory [61,62] and Grice's [16] maxim of relation, which allow us to view expressions of sentiment as purposeful communicative acts, and therefore justifying classification of sentiments regardless of the opinion holder of the expressed sentiment. For instance, even when the speaker is only describing someone else's sentiment (e.g., 'my girlfriend loved it'), they are doing this for communicative purposes, such as providing sentiment-relevant information in the absence of first-hand experience. With this in mind, our instructions (see S2 File for detailed instructions in full) include clarifications for each of the nine sentence types, such as:

**Contradicting sentiments towards different targets:** When a sentence contains both positive and negative expressions of sentiment towards different targets, the sentiment label should be based on the strength of the sentiment towards each target. Equally strong sentiments result in a 'mixed' label: "I liked the food, but the waiter was rude" (mixed). A clearly stronger sentiment towards one target overrides the sentiment towards the other target/s: "I liked the food, but the waiter was very rude" (negative). Examples:

- The food was excellent, although the service was slow (positive)

- I like the colour, but I hate how small the buttons are (negative)

- He likes the colour, but dislikes the design (mixed)

Finally, the detailed instructions were summarised as follows:

Neutral sentiment should only be chosen when there is no indication of sentiment in the text. Mixed sentiment should be chosen when both positive and negative sentiment are expressed and these are equally strong (even if the sentiments are expressed towards different targets, and even if the sentiments are expressed by different opinion holders). Positive or negative sentiment should be chosen if it is explicit (even when other sentiments are implied), if it is stronger than other sentiments expressed in the sentence, if it is the intended meaning of a sarcastic sentence, a rhetoric

question, or a request, or if it is inferable from the explicit or sentiment-expressing words in perspective-dependent sentences.

After reading the instructions (detailed or minimal), the participants were asked to demonstrate whether they had read the instructions properly by labelling seven practice items. This was to allow some familiarisation with the task, and also to encourage participants to read the instructions properly if they had not done so already. The practice items were unanimously deemed as acceptable by three native English speakers during the stimuli preparation but were not included in the final dataset to be annotated as the main task of the study.

### Annotation dataset

The AmbiSent dataset consists of 252 sentences with ambiguous sentiment, such that several plausible annotation choices could be made when annotating most of these sentences. For instance, the sentences may contain conflicting sentiments from several opinion holders, or the sentiment may depend on the perspective taken when interpreting the sentence. The possible labels for the text samples were Positive, Negative, Mixed, or Neutral. Nine different types of ambiguous sentences were constructed, with an example for each depicted in Table 1. There were seven sentences per sentiment label within each sentence type (based on the initial ground truth). In session 1, four attention check items were presented intermixed with the sentences, such that only one of the sentiment labels demonstrated appropriate attention paid to the task (e.g., 'Please choose the negative label to show you pay attention to the task').

### Sentiment judgment questionnaire

The questionnaire consisted of three parts. Part 1 contained two open-ended questions, with a minimum of 30 words (150 characters) required as a response for each. These questions gauged the overall experience and the general approach taken to complete the task:

• **Describe your experience completing the labelling task**, during both sessions. For example, describe what was easy and what was difficult about the task, and any feedback on how to improve the task.

• **Describe your process of choosing a sentiment label during the task** (both sessions). For example, when you chose a sentiment label for a sentence, what things did you pay attention to in the sentence? What were your reasons or criteria for choosing a sentiment label for different sentences and did you apply these criteria consistently?

Part 2 consisted of 15 close-ended questions, with a 5-point, fully labelled Likert-scale response format [63], probing the general experience and specific challenges in the annotation task, such as 'I found the task difficult' or 'The task

**Table 1. Example sentences and labels for each sentence type.**

| Sentence type | Example sentence | Label |
|---|---|---|
| Appraisal-dependent | The war has created millions of refugees. | Negative |
| Explicit vs Appraisal | I'm relieved that they fired Jane. | Positive |
| Explicit | I like warm summer evenings. | Positive |
| Perspective-dependent | England lost to France. | Negative |
| Rhetorical questions | Why must we argue all the time? | Negative |
| Sarcasm | I love being criticised first thing in the morning! | Negative |
| Supplications and requests | May God help those displaced by war! | Positive |
| Two opinion holders | I liked it, but my girlfriend disliked it. | Mixed |
| Two targets | I liked the food, but the waiter was rude. | Mixed |

instructions were too short, so I wasn't always sure how to do the task'. In addition to the 15 questions, one attention check question was included, where only the response options 'Strongly disagree' or 'Disagree' indicated an appropriate attention paid to the task.

Part 3 consisted of 18 close-ended questions, probing different aspects of the text that annotators might utilise when making annotation choices. These questions consisted of two examples for each of the nine sentence types, and the response options included a sentiment label and a potential reason for choosing this label, such as:

What was the sentiment label and your reason for the label you chose for sentences such as 'I thought the concert was great, but my friend found it too noisy.'?

a) Positive, because the sentiment of the speaker was positive

b) Negative, because the described situation is undesirable and likely leads to a negative sentiment

c) Mixed, because equally strong positive and negative sentiments were expressed

d) Neutral, because there were both positive and negative words that expressed equally strong sentiment

e) Other reason, please specify the label and your reason for it

The sentences used in Part 3 were all presented only in session 1 during the main annotation task. In addition to the 18 questions, one attention check question was also included, where only one of the response options indicated an appropriate attention paid to the task ('c) Mixed, because this was the only option that allowed me to demonstrate I paid attention to this question'). However, the free-text response was evaluated for any participant choosing the option 'e)' for the attention check question.

## Analysis

A mixed-methods approach was employed to draw insights from both quantitative and qualitative data collected during the study. Quantitative measures were used to assess inter- and intra-annotator agreement across the instruction groups, as well as common challenges in the annotation task, while a thematic analysis of annotators' free-text responses was conducted to explore their annotation strategies, challenges, and experiences. The qualitative component served not only to identify recurring themes in annotator experience, but also to help explain unexpected patterns in the quantitative results (see Reflexive thematic analysis of annotation experiences).

## Demographic characteristics questionnaire

Responses to the questionnaire were summarised both across and within instruction groups. Categorical data were reported as participant counts, and central tendencies were computed for numeric variables such as age and socioeconomic status. As the questionnaire required a response to each item before proceeding, there were no missing data. For free-text entries under 'Other' response options, each unique response was treated as a separate category, and its frequency was calculated in the same way as predefined response options.

## Annotation task

For both instruction groups, the following measures were computed:

- **H1:** Overall inter-annotator agreement for the full set of items annotated in the first testing session, quantified as Krippendorff's alpha [64]. This measure is suitable for the current setting, where all annotators annotate all items and agreement is estimated for nominal data [65]. The comparison between instruction groups was performed using a permutation test, with 5,000 permutations and a significance level of .05 [66].

- **H2:** Intra-annotator agreement for each annotator, based on their annotation of the 72 repeating items (i.e., items that were presented in both testing sessions). The intra-annotator agreement was quantified as the proportion of items that received the same sentiment label in both testing sessions, out of all the items that were presented in both sessions. The mean intra-annotator agreement was compared between instruction groups, using independent samples t-test.

- **RQ2:** Inter-annotator agreement for each item group (all items annotated in the first testing session), quantified as Krippendorff's alpha. This exploratory investigation was based on a descriptive comparison of annotator agreement across different item groups.

### Sentiment judgment questionnaire

Reflexive thematic analysis was applied to the free-text responses for the two open-ended questions in the questionnaire (Part 1 of the questionnaire). This analysis was informed by Braun and Clarke [67,68], with the aim to uncover prevalent themes in the annotation experience, as described by the annotators. The first author (LA) performed this analysis, treating each instruction group as a separate source of data. The analysis was conducted independently for each group to allow patterns of meaning to be identified within their specific context. After developing themes separately, similarities and differences across groups were considered.

For each Likert-scale question (Part 2 of the questionnaire), median responses were calculated. The most challenging aspects of the annotation task were identified as Likert-scale statements with highest medians, for all participants and by instruction group. The general and by-group medians were also visualised, to facilitate identifying particularly challenging aspects of the annotation task experienced by both groups, as well as any challenges experienced mostly by one of the groups (i.e., challenges dependent on the instructions received).

Both of these analyses (thematic analysis and the descriptive comparisons of responses to Likert-scale questions) were used to answer the **RQ1**. The thematic analysis was also used to help explain the quantitative results (H1, H2, RQ1, and RQ2).

The responses to questions regarding each item type (Part 3 of the questionnaire) were summarised by instruction group, via entropy H-values [69], reflecting the diversity of the responses. Higher H-values indicate a more even distribution across response options (i.e., greater response diversity), whereas lower values suggest consensus or convergence on a single option. An H-value of 0 reflects total consensus, where all responses fall into one category. The theoretical maximum entropy is achieved when all response options are equally probable, and this upper bound increases with the number of distinct options available. The H-value was calculated based on the percentages of annotators who chose each of the response options. When the 'other' response option was chosen, this was included as a unique option for the purposes of calculating the H-value. The H-values from both groups were compared using a permutation test, with 5,000 permutations and significance level of .05. This non-parametric method was chosen as it is suitable for small samples and non-normally distributed data. However, due to the small number of questions included, these comparisons are best seen as a tentative indication of whether response diversity varies by types of instructions received. The response diversity between the groups was also visualised. This exploratory analysis was used to answer the **RQ3**.

### Power analysis

The choice of sample size for annotators and for the number of items was based on a pilot study, where we tested a sample of 216 items and 252 items, each sample annotated by 10 participants (five participants receiving detailed instructions and five participants receiving minimal instructions). The pilot served two purposes: 1) to inform the choice of item sample, based on two criteria described below, and 2) to aid in estimating the required annotator sample size, by informing the power analyses.

The item sample size was determined by considering two criteria: 1) the duration of each testing session, which should not exceed two hours, as task performance tends to deteriorate with prolonged task engagement [70,71], and 2) the mean proportion of correctly annotated items in the first session, assessed against the initial ground truth labels based on the detailed instructions. Based on these two criteria, the results from the pilot data with 252 items were not considerably different than those from the pilot data with 216 items. Hence, we opted for proceeding with the 252 items as the final sample size (see OSF for detailed results from the pilot studies), to allow for the creation of a larger dataset.

The required annotator sample size was estimated via two a priori power analyses, one for each of our hypotheses, informed by the pilot data (252 items).

For H1 regarding inter-rater agreement, the required sample size was estimated via a simulation-based power analysis. More specifically, annotator data was simulated based on the sentiment label distribution for each item in the pilot data (where the mean difference of Krippendorff's alpha between the groups was 0.14). As our pilot data was relatively small (five annotators per instruction group), we applied additive Laplace smoothing (with smoothing factor 0.2) to ensure non-zero probabilities for response options (sentiment labels) not chosen in our pilot data for some of the items. This was done to provide more realistic simulation of the data, via more equiprobable sentiment choices, while still basing the sentiment choice probabilities on the pilot data. The simulations were run with increasing sample size, starting from 10 annotators per group, with 1,000 sample pairs simulated for each sample size. Permutation tests were conducted for each simulated sample pair (comparing the difference in Krippendorff's alpha between the two groups), and the statistical power was calculated as the proportion of statistically significant permutation test results (at power level of 0.80 and alpha threshold of 0.05) out of 1,000 simulated sample pairs. Based on this procedure, the required sample size for H1 was 10 participants per group, as the statistical power was already 0.99 with this sample size.

For H2 regarding intra-annotator agreement, the pwr package in R [72] was used to conduct a power analysis for independent samples t-test (one-tailed), based on means and pooled standard deviation from the pilot data. With power of 0.80 and alpha level of 0.05, the required sample size per group was 53 participants, allowing detection of effect size of 0.49 (Cohen's *d*). Given the constraints of our small sample size, this power calculation should be viewed as a cautious estimate. However, in the absence of more robust data specific to this type of intra-rater reliability measures, we used this estimate to guide the recruitment for the study.

Based on these two power analyses, we recruited 53 participants per group. Apart from sufficiently powered study design, this sample size should provide a representative distribution of ratings, and thus reliable soft labels for the dataset. Data quality was ensured by evaluating attention checks prior to participant inclusion, where failed participants were replaced with newly recruited participants until the desired sample size was reached (see Exclusion of participants). Note, this process involved analysing the attention checks only, not the main data, thus avoiding repeated hypothesis testing.

## Exclusion of participants

Participant engagement is crucial for investigating the influence of instructions on annotator agreement, as only participants who understand and follow the detailed instructions appropriately can demonstrate the intended effects of the experimental manipulation (i.e., higher inter-rater and intra-rater agreement compared to the Minimal group). Piloting the study raised a concern of low participant engagement, which is why we chose to increase the required number of attention checks participants would need to pass to be included. Instead of three out of six attention checks correct across sessions (required in our pre-registration before the pilot study), all participants were required to pass five out of six attention checks (four in session 1 and two in session 2) to be included in the analyses. The requirement to pass five rather than all attention checks balanced the desired data quality with practical feasibility, as requiring perfect scores would have excluded about half of the participants, based on the pilot data. With this requirement, we continued data collection until we had 53 participants per group, counting the first 53 who successfully completed both sessions. Out of 145 participants who started the study, 17 abandoned it before giving initial consent (i.e., decided not to participate), and were therefore

not compensated. Out of the remaining 128 individuals successfully recruited, 22 were excluded for the following reasons: nine participants due to failing attention checks, 10 participants due to not completing session 2, and three participants due to over-recruitment. While exclusion due to failed attention checks or incomplete data was essential for maintaining data integrity, we recognize the ethical responsibility to ensure fair treatment of all participants. Accordingly, excluded participants were compensated pro rata for their time spent on the task, ensuring no participant was disadvantaged financially. Over-recruited participants were compensated on the same basis as included participants (see section Data collection and compensation).

### Analysis software

The thematic analysis was conducted in NVivo (version 14). All other analyses were conducted in R (version 4.4.3), either using custom scripts and base R functionality, or by employing specific packages, such as irrCAC package [73] for Krippendorff's alpha, and pwr package [72] for power analysis for H2.

## Results

The data collection for this study took place between the 19th of March and the 3rd of April 2025. Eight participants were excluded for failing attention checks and 10 participants were excluded for not completing the second session. New participants were recruited until both groups (Detailed and Minimal) consisted of 53 participants that had successfully completed both sessions.

Below, the sections from Participant demographics to Response diversity present the results for the quantitative strand of the study, the section Reflexive thematic analysis of annotation experiences consists of the qualitative strand, and the integration of the findings from both strands is presented in the section Integration of quantitative and qualitative findings.

### Participant demographics

Table 2 depicts a summary of the participant demographics overall and by group (Detailed or Minimal). Although quota sampling was used to achieve even numbers of female and male participants, this was not achieved perfectly, due to participant attrition (e.g., non-consent, time-outs or study abandonment, all of which led the automated recruitment system to continue as if these participants had completed the study).

Overall, the participants were on average in their mid-30s, and approximately half of them were female, white, native speakers of English from England, and had completed a university-level education. All of the demographic characteristics tracked here were considerably evenly distributed across the instruction groups.

### Participants' relevant task experience

A summary of the participants' experience with data annotation and sentiment analysis is provided in Table 3. Overall, the level of experience in both was evenly distributed across the two groups. Over half of the participants had some experience with data annotation tasks. Over half of the participants had a vague idea of what sentiment analysis was, or had never heard of it before taking part in the study.

### Task duration and correct responses

The time spent on the tasks and the proportion of correct responses (against the initial ground truth from at least three native English speakers) can be used as indications of participant engagement. These are summarised in Table 4, for each participant group. As seen in this table, the Detailed group consistently spent more time on the tasks and achieved a higher proportion of correct responses against the initial ground truth. Moreover, each participant scored above 40% correct, exceeding the 25% chance level for a four-option task, indicating reasonable engagement and comprehension of

**Table 2. Participant demographics by instruction group.**

| Age | Full sample (n = 106) | Detailed (n = 53) | Minimal (n = 53) |
|---|---|---|---|
| Mean | 34.65 | 34.87 | 34.43 |
| SD | 10.40 | 10.51 | 10.38 |
| Min | 19 | 20 | 19 |
| Max | 63 | 62 | 63 |
| **Gender** | | | |
| Man | 49 | 29 | 20 |
| Non-binary | 1 | 1 | 0 |
| Woman | 56 | 23 | 33 |
| **Education (Highest level completed)** | | | |
| Lower secondary education | 2 | 2 | 0 |
| Upper secondary education | 13 | 6 | 7 |
| Post-secondary non-tertiary education | 6 | 4 | 2 |
| Short-cycle tertiary education | 8 | 3 | 5 |
| Bachelor's or equivalent level | 50 | 22 | 28 |
| Master's or equivalent level | 19 | 13 | 6 |
| Doctoral or equivalent level | 8 | 3 | 5 |
| **Race or Ethnic group** | | | |
| American Indian or Alaska Native | 1 | 1 | 0 |
| Asian | 4 | 3 | 1 |
| Black, Caribbean or African | 38 | 21 | 17 |
| Hispanic or Latino | 4 | 1 | 3 |
| Mixed or Multiple ethnic groups | 4 | 1 | 3 |
| Other: South African | 1 | 0 | 1 |
| White | 54 | 26 | 28 |
| **Sex** | | | |
| Female | 57 | 24 | 33 |
| Male | 49 | 29 | 20 |
| **Socioeconomic status (self-assessed: 1–10)** | | | |
| Mean | 5.52 | 5.36 | 5.68 |
| SD | 1.62 | 1.63 | 1.61 |
| Min | 1 | 1 | 2 |
| Max | 9 | 8 | 9 |
| **Variant of English spoken natively** | | | |
| Australia | 3 | 1 | 2 |
| Canada | 5 | 2 | 3 |
| England | 58 | 31 | 27 |
| Other: British | 1 | 1 | 0 |
| Other: Ireland | 2 | 1 | 1 |
| Other: New Zealand | 3 | 1 | 2 |
| Other: South Africa | 16 | 7 | 9 |
| Other: United Kingdom | 1 | 1 | 0 |
| United States | 17 | 8 | 9 |

*Note.* The numbers in the table reflect counts of participants choosing a particular category, unless otherwise specified (for Age and Socioeconomic status). The demographics are shown for the full sample of participants (n = 106), as well as separately for the Detailed group (n = 53) and the Minimal group (n = 53).

Table 3. Participants' experience with data annotation and sentiment analysis.

| Experience with data annotation tasks | Full sample (n=106) | Detailed (n=53) | Minimal (n=53) |
|---|---|---|---|
| I have no experience | 22 | 12 | 10 |
| I have some experience | 73 | 37 | 36 |
| I have extensive experience | 11 | 4 | 7 |
| **Experience with sentiment analysis** | | | |
| I had never heard of it before | 45 | 21 | 24 |
| I had a vague idea of what it was | 22 | 13 | 9 |
| I knew what it was, but haven't conducted sentiment analysis | 17 | 7 | 10 |
| I have some experience conducting sentiment analysis | 16 | 9 | 7 |
| I have extensive experience conducting sentiment analysis | 6 | 3 | 3 |

*Note.* The statements are shortened for brevity, see the complete statements in the demographic questionnaire (S1 File).

Table 4. Session durations and proportions of correct responses by instruction group.

| Session 1 duration (minutes) | Detailed | Minimal |
|---|---|---|
| Mean | 67.90 | 51.52 |
| Median | 61.60 | 41.08 |
| SD | 36.31 | 49.18 |
| Min | 16.83 | 14.77 |
| Max | 172.33 | 363.70 |
| **Session 2 duration (minutes)** | **Detailed** | **Minimal** |
| Mean | 46.63 | 34.92 |
| Median | 38.97 | 30.92 |
| SD | 30.69 | 20.01 |
| Min | 9.32 | 10.15 |
| Max | 199.42 | 98.33 |
| **Proportion correct** | **Detailed** | **Minimal** |
| Mean | 0.70 | 0.67 |
| Median | 0.71 | 0.68 |
| SD | 0.10 | 0.07 |
| Min | 0.41 | 0.46 |
| Max | 0.86 | 0.80 |

the instructions. However, the mean difference in the correct responses between the instructions groups was small, which may suggest low task engagement by participants in the Detailed group, or that the Minimal group spontaneously annotated the sentences in line with the detailed instructions.

The correct responses were also quantified as the majority vote for each sentence. When quantified this way, the Detailed group achieved a higher proportion of correct responses against the ground truth overall, and for most sentence types (Table 5), except for Appraisal and Rhetorical sentence types.

### Inter-rater reliabilities in session 1

Inter-rater reliabilities were quantified as Krippendorff's alpha (α), which can take values between −1 (perfect disagreement) to 1 (perfect agreement), with 0 indicating no agreement beyond chance. The inter-rater reliabilities for the full dataset of 252 sentences were α=0.50, 95% CI [0.47, 0.53] in the Detailed group and α=0.59, 95% CI [0.56, 0.63] in the Minimal

**Table 5. The proportions of correct responses as majority vote by sentence type and group.**

| Sentence type | Detailed | Minimal |
|---|---|---|
| Appraisal | 0.93 | 0.96 |
| Explicit vs Appraisal | 0.96 | 0.93 |
| Explicit | 1.00 | 1.00 |
| Perspective | 0.93 | 0.75 |
| Rhetorical | 0.89 | 0.96 |
| Sarcasm | 1.00 | 0.86 |
| Supp & Requests | 0.71 | 0.68 |
| Two opinion holders | 0.68 | 0.50 |
| Two targets | 0.64 | 0.50 |
| **All sentences** | **0.86** | **0.79** |

group. The difference of 0.09 between the inter-rater reliabilities is thus in the opposite direction to our hypothesis (H1). A 2-tailed permutation test with 5000 permutations resulted in a statistically significant difference between the inter-rater reliabilities across the instruction groups (p < .01), and remains significant under a Bonferroni correction (α = 0.01, for five tests across the study). This unexpected result will be explored further in the section 'Integration of quantitative and qualitative findings'. It is worth pointing out that commonly cited thresholds for Krippendorff's alpha are values between 0.667 and 0.800 for tentative conclusions, whereas over 0.800 are generally considered reliable [74] (p. 241–242). Our findings reveal low inter-annotator reliability overall, which likely reflects inherent ambiguity in sentiment judgments, but may also be indicative of inconsistencies in instruction adherence. However, the actual reliability thresholds should be considered based on the intended purpose, as pointed out by Krippendorff [74] and others [75,76]. Furthermore, inter-rater reliabilities tend to decrease with increasing rater sample sizes [28,77] as more diverse viewpoints introduce greater potential for disagreement. Thus, our reliability estimates are not simply indicative of low annotation agreement, but also a natural consequence of the large number of annotators involved in our study. This is in contrast to studies with only two or a handful of annotators per item, which may artificially inflate reliability estimates by underrepresenting the true range of interpretations.

To answer RQ2, the inter-rater reliabilities were inspected descriptively for different sentence types (no formal statistical comparisons were conducted due to small number of sentences per sentence type). Fig 2 depicts the inter-rater reliabilities for each sentence type, across participants and by instruction group.

Overall, particularly challenging sentence types, regardless of the instructions received, were Perspective, Sarcasm, Rhetorical and Supp & Request sentences. Each of these sentence types resulted in inter-rater agreements just at or below 0.50, in both groups, with Supp & Request sentences as low as 0.27 (Detailed group) and 0.31 (Minimal group). The only clear numerical differences between the instructions groups were seen in agreement levels for Appraisal and especially for Two opinion holders and Two targets sentence types. For these sentences, the Minimal group consistently achieved an inter-rater agreement above 0.60, whereas the Detailed group's agreement remained just above 0.50 for Appraisal sentences, and just below 0.40 for the Two opinion holders and Two targets sentences. Overall, the agreement in the Minimal group was higher for most sentence types, except for Sarcasm, where the agreement between the groups was equal (both at 0.5), and Perspective and Rhetorical sentences, where the Detailed group achieved slightly higher agreement (both at 0.48, compared to both at .046 in the Minimal group).

### Intra-rater reliabilities in session 2

Participants completed the second session seven days after completing the first session. The inter-session interval ranged from 6.4 days to 8.4 days, with 1st and 3rd quartiles 6.7 and 7.0, respectively. The variation around the intended seven-day inter-session interval was comparable between the Detailed and the Minimal groups.

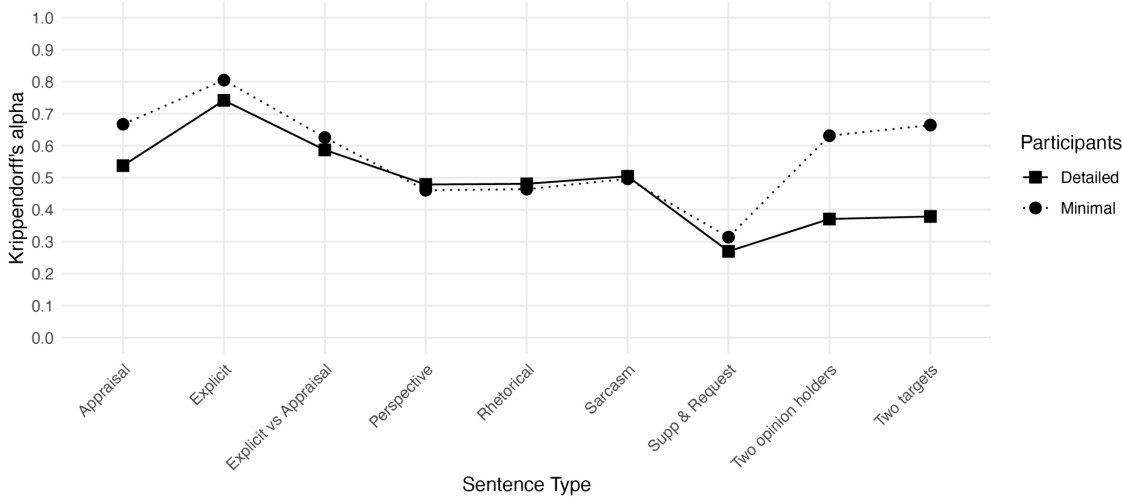

**Fig 2. Inter-rater reliabilities for different sentence types.**

The mean percent intra-rater reliabilities for the 72 repeating items across sessions were compared between the groups. Against our hypothesis (H2), the intra-rater agreement was lower in the Detailed group ($M = 76.81$, $SD = 11.97$) than in the Minimal group ($M = 80.21$, $SD = 9.29$), and no statistically significant difference was found between the groups in the expected direction: $t(104) = 1.64$, $p = .95$, 1-tailed, Cohen's $d = -0.32$. However, the difference between the groups in the opposite direction approached statistical significance ($t(104) = 1.64$, $p = .052$, 1-tailed, Cohen's $d = 0.32$). Although our study was not powered to detect differences in this direction (Minimal > Detailed) due to our directional hypothesis, this trend warrants further exploration (see section 'Integration of quantitative and qualitative findings').

Inspecting the mean percent intra-rater reliabilities by sentence type (Table 6), the Minimal group maintains higher reliabilities for most sentences, particularly for Two opinion holders and Two targets. However, the Detailed group annotated Rhetorical sentences slightly more consistently across sessions.

**Table 6. Percent intra-rater reliabilities by instruction group and sentence type.**

| | Detailed group | | | | Minimal group | | | |
|---|---|---|---|---|---|---|---|---|
| Sentence type | Mean | SD | Min | Max | Mean | SD | Min | Max |
| Appraisal | 76.42 | 19.09 | 37.5 | 100 | 80.66 | 16.91 | 12.5 | 100 |
| Explicit | 87.5 | 12.01 | 62.5 | 100 | 89.15 | 13.66 | 25 | 100 |
| Explicit vs Appraisal | 80.42 | 15.6 | 25 | 100 | 80.9 | 16 | 25 | 100 |
| Perspective | 71.7 | 20.09 | 12.5 | 100 | 72.88 | 16.4 | 25 | 100 |
| Rhetorical | 74.76 | 18.26 | 25 | 100 | 71.7 | 16.47 | 12.5 | 100 |
| Sarcasm | 80.42 | 17.59 | 37.5 | 100 | 82.08 | 15.41 | 25 | 100 |
| Supp & Request | 62.5 | 19.46 | 12.5 | 100 | 64.86 | 17.69 | 37.5 | 100 |
| Two opinion holders | 79.95 | 18.4 | 37.5 | 100 | 90.8 | 15.34 | 25 | 100 |
| Two targets | 77.59 | 19.05 | 37.5 | 100 | 88.92 | 15.04 | 12.5 | 100 |
| **All Sentences** | **76.81** | **11.97** | **48.61** | **97.22** | **80.21** | **9.29** | **44.44** | **91.67** |

## General challenges in annotation

To answer RQ1, general challenges in the annotation task are summarised as median scores to close-ended questions from both groups. Fig 3 depicts the combined responses from both groups, Fig 4 the responses from the Detailed group, and Fig 5 the responses from the Minimal group. For brevity, these figures contain short versions of the statements in the questionnaire (see full versions in S3 File).

As seen in these figures, the Detailed group mostly disagreed or chose the neutral option for the statements. The participants generally agreed that insufficient context made the annotation task more difficult (median response 'Agree'). They also strongly disagreed with the idea that the instructions were too short (median response 'Strongly disagree').

The Minimal group agreed with more items. The participants agreed that insufficient context, not knowing how much to interpret the text, and uncertainty about sarcasm made the task more difficult (median response: 'Agree'). They also indicated a tendency to interpret implied sentiment in text (median response: 'Agree'). The only items that were strongly disagreed with had to do with instructions – the participants did not check the instructions often during the task, and they did not think that the instructions were too short nor too long (median response: 'Strongly disagree').

The participant responses are also depicted in more detail in Fig 6 (Detailed group) and Fig 7 (Minimal group). Several interesting observations can be made from these figures, but we will focus only on the key observations that are relevant

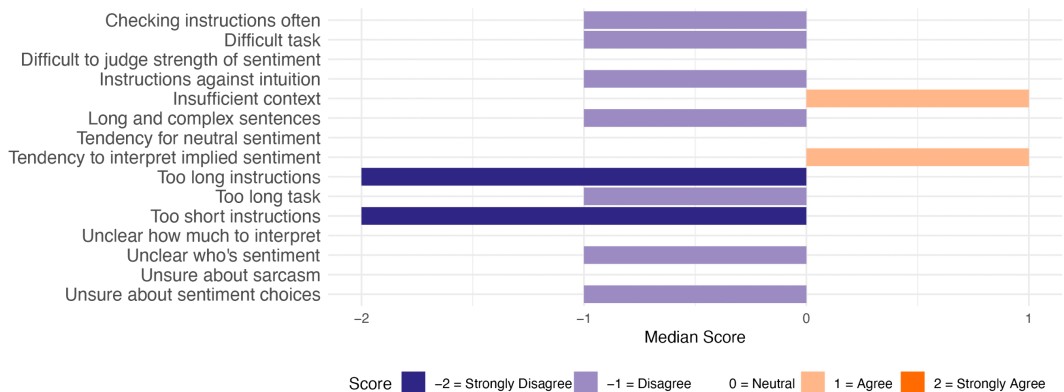

**Fig 3. Median responses to General Challenges statements.** Responses from all participants (n = 106).

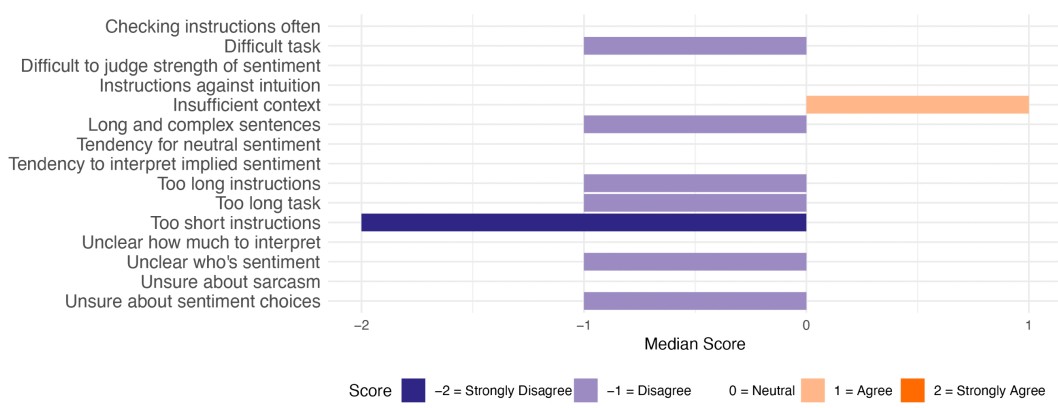

**Fig 4. Median responses to General Challenges statements.** Responses from Detailed group (n = 53).

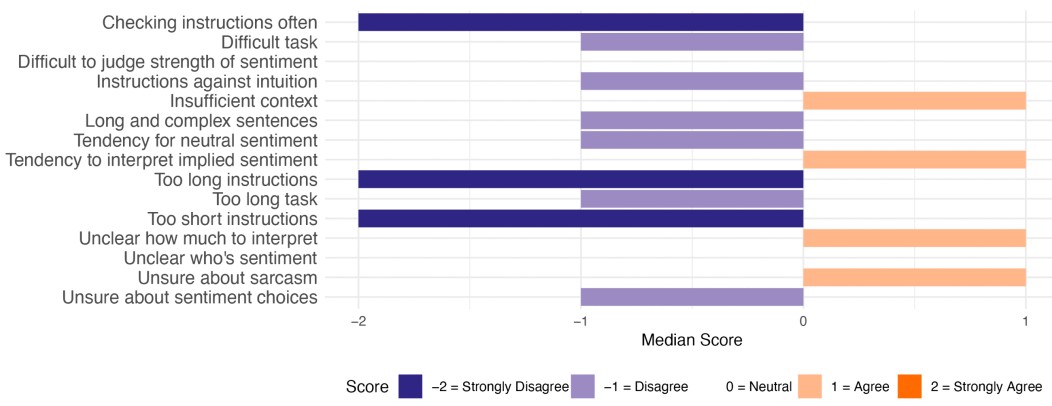

**Fig 5. Median responses to General Challenges statements.** Responses from Minimal group (n = 53).

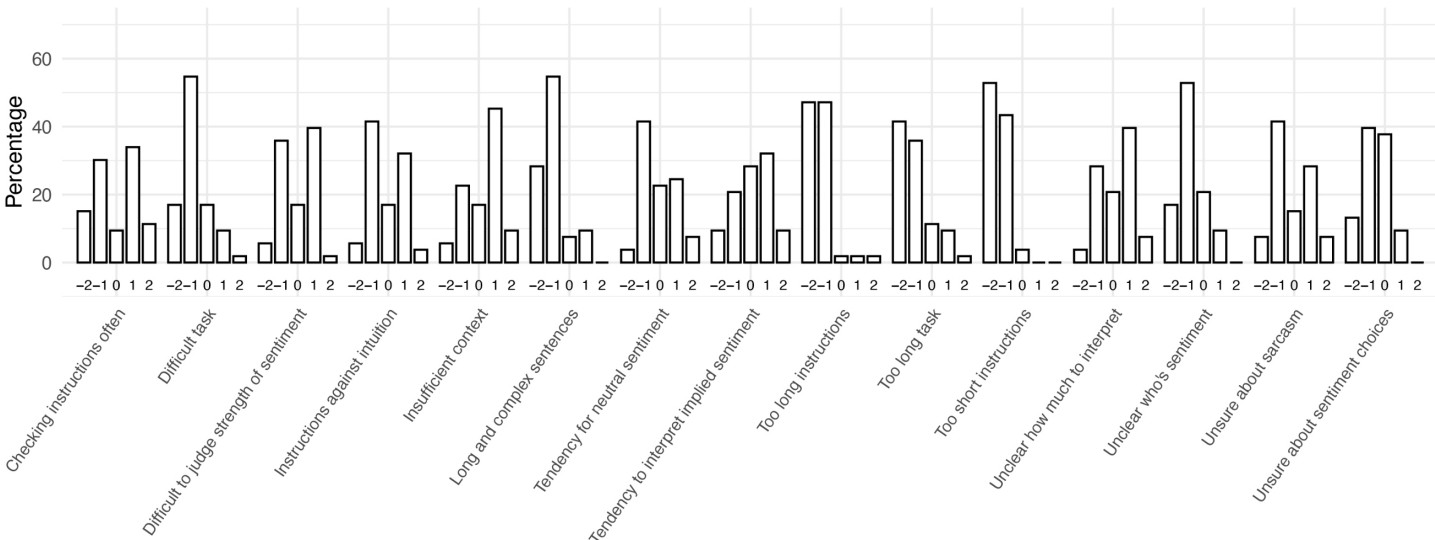

**Fig 6. Detailed group's distribution of chosen response options for General Challenges statements.** −2 = Strongly disagree; −1 = Disagree; 0 = Neither agree nor disagree; 1 = Agree; 2 = Strongly agree.

for integrating the results with the qualitative analysis. Firstly, the two statements relating to following instructions (i.e., checking instructions often, and instructions being against intuition) were broadly disagreed on by the Minimal group (94% and 64% of participants chose 'Strongly disagree' or 'Disagree' for these, respectively). By contrast, the Detailed group was more divided regarding these statements (45% and 47% of participants chose 'Strongly disagree' or 'Disagree' for these, respectively, while 45% and 36% of participants chose 'Strongly agree' or 'Agree' for these, respectively).

Secondly, sarcasm was considered more of a challenge by the Minimal group (58% of participants choosing 'Agree' or 'Strongly agree', while 23% broadly disagreed with the statement). The Detailed group was more divided about this statement (36% broadly agreeing with this statement while 49% broadly disagreed).

## Response diversity

To answer RQ3, diversity of reasons for annotation choices was quantified as Shannon's entropy H and the mean entropy H between groups were compared with a permutation test. As outlined in the pre-registration, the 'other' responses were

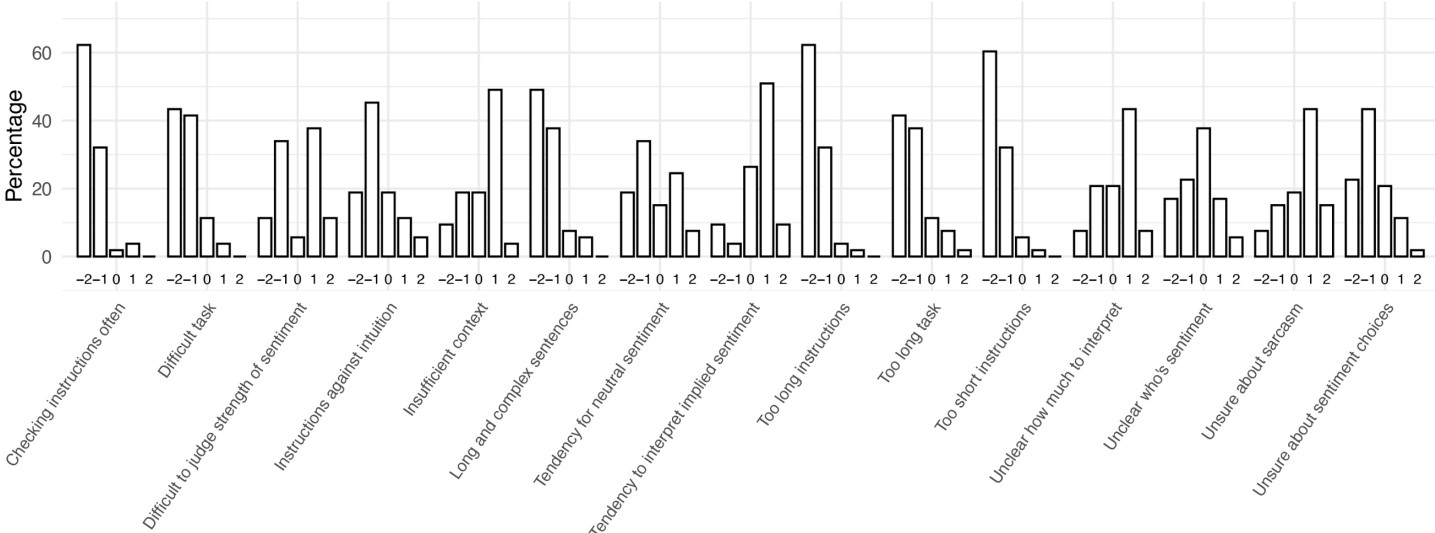

**Fig 7. Minimal group's distribution of chosen response options for General Challenges statements.** −2 = Strongly disagree; −1 = Disagree; 0 = Neither agree nor disagree; 1 = Agree; 2 = Strongly agree.

each counted as a unique response for a given statement. The mean entropy H were very similar in both groups: 0.90 (SD = 0.45) in the Detailed group and 0.89 (SD = 0.51) in the Minimal group. As such, the Minimal group gave slightly less diverse reasons for annotation choices than the Detailed group, but this difference was not statistically significant. The observed difference was Δ = −0.01 (Minimal – Detailed), p = 0.94, with 5000 permutations.

Entropy H values for each sentence type (two example sentences per sentence type) are depicted in Fig 8. The Minimal group gave consistently more diverse reasons for their annotation choices for Appraisal, Explicit vs Appraisal, and Sarcasm sentences, whereas the Detailed group gave consistently more diverse reasons for annotations for Supp & Requests and Two targets sentences. It should be noted, however, that the low number of sentences in this part of the questionnaire renders these comparisons merely illustrative rather than definitive.

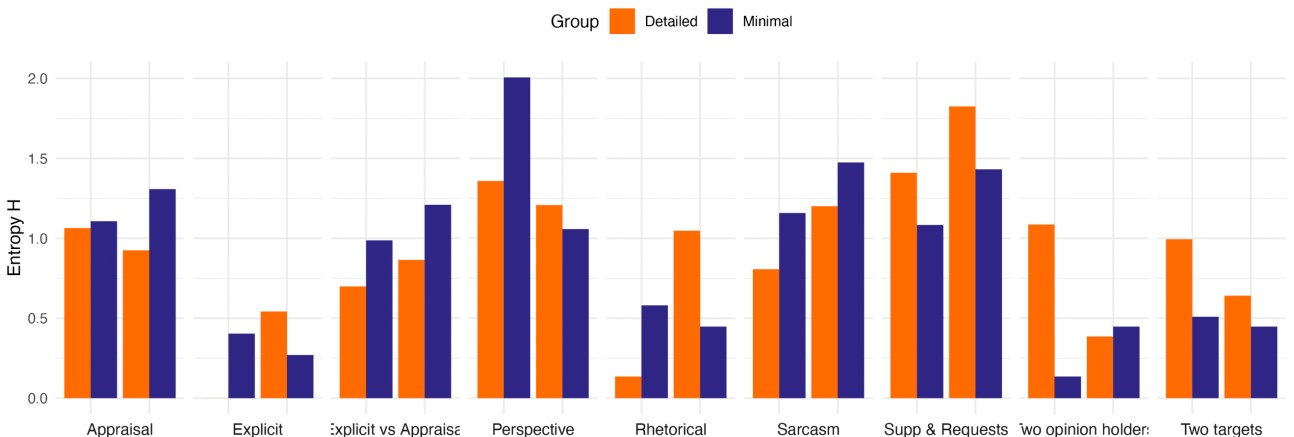

**Fig 8. Entropy H values indicating diversity of reasons given for annotation choices by Sentence type and Instruction Group.** Two example sentences were presented per Sentence Type.

### Reflexive thematic analysis of annotation experiences

The reflexive thematic analysis [67,68] was originally designed to serve as a source of information about the challenges the participants experienced, in their own words, *before* they were exposed to our framing of the annotation process. However, given the surprising findings regarding the inter-rater reliabilities across instruction groups, the qualitative analysis may provide useful insights more broadly, which could help explain and contextualize the unexpected quantitative findings. As such, we extended the scope of the thematic analysis to include a more open-ended exploration of participant experiences.

This *post hoc* expansion makes our study a hybrid between convergent and sequential mixed-methodological designs. Although the study followed a convergent mixed-methods design in terms of data collection — with quantitative and qualitative data gathered concurrently — the analysis was guided by an explanatory logic: after analysing the quantitative data, we identified unexpected patterns, which led us to broaden the scope of the qualitative analysis to explore possible explanations for these findings. In this sense, the qualitative data served a secondary, explanatory function, consistent with explanatory sequential design. The integration of results from both types of data was achieved through merging, with quantitative and qualitative findings brought together at the interpretation stage using a joint display [78]. While this was not part of the original pre-registered plan, it is a pragmatic response to emergent data patterns. Furthermore, as the open-ended questions probed the general experience during the annotation, the generated free-text responses should be readily suitable for our extended purposes.

### Researcher positionality and theoretical orientation

The analysis was conducted by the first author, a postdoctoral researcher with a background primarily in quantitative methods, supplemented by training and experience in a few qualitative projects. The researcher's disciplinary background is in psychology and cognitive science, with experience in both clinical and research contexts.

The researcher adopted a pragmatist epistemological stance, whereby analytic utility and the goal of understanding participants' annotation experiences in relation to the quantitative findings was prioritised. This stance was combined with a semantic and experiential interpretation of the data, treating participants' language as a reflection of their lived annotation experiences. While coding was open and data-driven, interpretation was informed by 1) seeking to understand the quantitative findings, and 2) an interest in the cognitive mechanisms underlying sentiment label decisions. Informed by Braun and Clarke's principles of reflexive thematic analysis [67,68], the researcher recognises their role in constructing themes via active engagement with the data.

Reflexive notes were maintained throughout the analysis to acknowledge and monitor any potential influence of prior expectations or disciplinary background. For instance, the researcher noted that analysing the Detailed group first likely resulted in stronger expectations for the themes that may emerge from the Minimal group. Nonetheless, the Minimal group's responses were approached with openness to identifying experiences unique to minimally instructed annotators.

### Researcher-participant relationship

As the responses were submitted anonymously through an online platform (Prolific), there was minimal direct interaction between the researcher and participants. Although a few participants sent clarification messages during the data collection stage, the researcher was not aware of which participants had made contact when coding the responses during the analysis stage.

### Data source

Each participant from both instruction groups wrote two free-text responses, each a minimum of 150 characters long. As such, a total of 212 free-text responses, 106 from each group, were analysed. Separate analyses were conducted for each group, as the different instructions were expected to result in distinct experiences and issues raised by the participants.

## Data description

While the nature of the data – relatively brief, focused reflections – limited the extent to which deep, interpretative themes could be developed, the method was still valuable in organising and making sense of common experiences and viewpoints. In this context, reflexive thematic analysis was used primarily to summarise and structure participants' insights, with an emphasis on how these related to the quantitative findings. This more descriptive use of reflexive thematic analysis was useful for mapping patterns across the dataset, rather than for interpreting latent meaning in depth.

Given the cognitively demanding nature of the open-ended questions, requiring a high level of introspection and the ability to articulate internal processes in writing, many responses were relatively limited in insight or irrelevant to the research question. Common patterns of uninformative responses included brief descriptions of the task itself, mentions of the order in which the task was performed, or general comments on the task's difficulty.

## Analytic procedure

1. **Familiarizing oneself with the data.** The dataset was read and re-read to gain an overall sense of annotator experiences, with a focus on annotator experiences that would be relevant in explaining the quantitative results, or informative about the annotator approaches taken. During this stage, notes were made on potential codes and interesting ideas to revisit or keep track of in the second reading.

2. **Generating initial codes.** The codes were developed more actively throughout the second reading of the dataset. Using NVivo (version 14) software, each comment was assigned a code if it contained something interesting relative to the analysis aims. If a comment consisted of several important ideas, it received a code for each. Against the researcher's natural tendency, the codes were kept as descriptive as possible, rather than condensed into higher-level concepts, to allow more interpretative freedom in the next phase of the analysis.

3. **Developing themes.** The initial themes were developed by combining codes that appeared meaningfully related, trying to condense them into a broader theme. This consisted of considerable naming and re-naming of themes, and changes in the initial hierarchy of themes and their potential sub-themes.

4. **Reviewing themes.** Further merging and re-structuring of themes and sub-themes took place. At this point, several themes could still have been condensed into a higher-level of abstraction, encompassing more general experience or pattern of meaning in the data. However, the researcher generally favoured higher level of detail over generality, to maximize their usefulness for the study aims. Each theme was inspected for coherence, by re-reading the coded data extracts and assessing whether they fit together neatly. The initial mind-map was created, for both instruction groups separately, and the comments were re-read for the purpose of evaluating the appropriateness of the themes relative to the entire dataset.

5. **Defining and naming themes.** The names and descriptions of each theme were reviewed. Some themes were recognized as more descriptive than interpretative (e.g., 'Challenging sentence types'), reflecting the nature of the data and the pragmatic focus on identifying annotation challenges. These were retained because they offered valuable summaries of participant-reported difficulties, despite being less interpretatively deep than is typical of reflexive thematic analysis.

6. **Producing the report.** Relevant data extracts were chosen to demonstrate how the data supports each theme. Due to the relatively large number of themes and sub-themes, that were deemed useful for detailed insights of the data, the analytical narrative was kept to a minimum. Instead, key findings relative to the study aims were discussed and linked with existing literature in a separate section (Discussion of thematic analysis findings).

## Detailed group

**Challenging aspects.** This theme encompasses the most challenging aspects of the annotation task, divided into four sub-themes.

**The volume of work** was considered challenging, including the length of the instructions and the volume of sentences to annotate, particularly in the first session. The difficulty was twofold: an initial sense of being overwhelmed by the task ahead, and the challenge of sustaining focus and mental effort over time.

> I found the task overall to be fairly straightforward after getting to study the instructions on the first page, which made the task seem more daunting at first. (P23)

> I also feel like the first session had a lot more sentences to label and it was much more overwhelming. Perhaps if I had kept going with more labeling in this session, I would have begun to lose focus. (P99)

> It was a fairly long repetitive task with quite small text, certainly for the longer first session so I took a few quick screen breaks during the task. (P81)

**Evaluating the strength of sentiment** was often described as challenging, particularly when modifying cues (e.g., 'very', 'somewhat') or clear opposites (e.g., 'love' vs. 'hate') were absent. Without these linguistic cues, participants found it harder to determine which sentiment was dominant.

> However, sometimes words were not antonyms, e.g., like versus unhappy so it was harder to tell which one dominated. (P38)

> I sometimes found it challenging to compare adjective in mixed expressions, especially ones without amplification. Eg does bored trump like? (P83)

> The most difficult ones to judge were the mixed positive/negative statements where I was unsure whether to consider one of them to be stronger than the other. (P57)

**Challenging sentence types** included Sarcasm and Supplications & Requests.

> The most challenging ones, I found, were statement such as "my husband did all the cleaning, what a terrible situation to be in" – this seems an overall positive sentiment, but using sarcasm so has negative terminology in it. (P18)

> it was also difficult to tell sarcasm, e.g., how many investment opportunities I presumed to be negative as no-one likes a lot of spam email. (P38)

> The only other ones I had trouble labelling were the "may we..." statements. (P78)

> I struggled most with the "supplication & request" expressions, especially the mixed ones. I found it difficult to assess if "prayers" and "thoughts" were in and of themselves positive enough to overcome or equal the rest of the sentence. (P83)

**Interpretation** encompasses several challenges in judging sentiment, highlighting the factors that influence how a piece of text is interpreted. This includes who the speaker is, the context of the scenario described in the text, whether and what kind of world knowledge is applied when reading the sentence, who is reading the sentence, and what state of mind the reader may be in.

> for example a sentence like "the shop was in the same spot as last time" is a bit difficult as it is dependent on the person reviewing. it can be either of the three options depending on who is reading it and in what context you want to take it in. (P64)

but some of the ones where sarcasm was (potentially) being used were a lot more difficult, for example "That tshirt looks so good on you, I love how big it makes your stomach look". I figured it was meant to be a negative but it sounded a little like something endearing my son might say to me, so I did have a sense that it could be meant in a sweet way. (P25)

But then came the tricky stuff. Sarcasm was a headache – like that sentence, "Oh great, another meeting!" Is it negative? Probably, but what if the person genuinely loves meetings? (Unlikely, but still.) (P95)

**Annotation approach.** This theme consists of different cognitive approaches to the annotation task, characterised by reliance on different aspects of the text samples when making annotation choices. Three sub-themes were identified.

**Word-focused approach**, in which sentences were scanned for sentiment-carrying words, and potentially for the number of words and their polarity. Mechanistic and potentially shallower processing appears characteristic to this approach.

The first thing I would look for is if the statement has any positive or negative word such as (e.g., love, dislike, exciting, etc). If the statement has no additional information I would mark it respectively based on what word was used. (P26)

I always paid most attention to the extremity of the positive/negative words. "Really love" would always be positive compared to just "hate," for example. I felt like that was what guided me through the labeling the most. If I didn't see any difference between the extremes of the words, it was mixed. If I didn't see any positive/negative words at all, it was neutral. (P99)

**From Word-focused to Holistic approach,** in which the initial word-focused screening of sentiment-carrying words is followed by more holistic interpretation of the overall emotional tone of the sentence, often based on intuitive contextual and linguistic understanding.

First, I focused on the words that actually implied whether it was positive or negative (such as love and hate) and decided which side it leaned towards based on the strength of the emotions they elicit. If it were still not clear then I would decide whether the sentence intended to give off a positive or negative response. Lastly I would use my gut to ask myself how i would feel about the subject of the sentence. (P84)

I looked for particular wording, for example, often when the statement used 'but', it fell into the mixed category (in my opinion). I choose to select positive/ negative depending on the overall judgement that that statement was making. (P53)

**Empathic, role-taking approach,** in which participants imagine themselves as the speaker (or receiver) of the sentence and evaluate the sentiment they would experience when producing (or receiving) the sentence.

I read the statements to myself and imagined what I would have felt in the moment if I were the one saying them. (P78)

I tried to feel it or try and see myself saying it and see which sentimental label it falls on. I had to imagine me talking to a friend or family member and see assume the reaction it will get. (P93)

As described by one respondent, the choice of the approach may also change depending on the sentence type.

For explicit statements, I simply paid attention to the sentiment words such as love or hate, very or little, etc. For things like appraisal dependent statements, I felt I had to put my mind in a different space and try to put myself in the shoes of the one making the statement, how I'd feel about it. (P23)

**Instructions** and their function in the annotation process were discussed from three distinct perspectives.

**Following instructions** was mentioned in the form of descriptions of the utility, comprehensiveness, or the ease of use of the instructions provided.

> I found the sentences which could have arguably been one label or another the hardest as I had to think about these the most and refer back to the instructions and examples given at the start of the task. (P63)

> I am glad that the instructions noted that the questions would save so I can freely go back to check the examples. (P84)

> I felt that the instructions were pretty thorough and covered most situations. While there were some edge cases where I was unsure, I was able to apply one or more or the rules to almost every example, making many of them easy. (P57)

**Reliance on intuition vs instructions** was described as a tension between the annotation instructions and how participants would intuitively annotate the sentences. This tension was often resolved by relying on intuition from the outset, or resorting to it when the instructions proved inconclusive.

> When deciding which sentiment label to select, I generally relied on my initial instinct and previous experience of positive/negative/neutral language use in everyday situations. (P48)

> I found if I really couldn't tell based on instructions I did tend to use my intuition to decide. (P25)

> I tried to approach the task linguistically according to the rules outlined at the beginning but would have found it easier to just use my instinct about what was positive or negative. However, it was difficult to judge the degrees of strength a word might have – by instinct I would have put these as mixed rather then positive if the positive emotion dominated. (P38)

**Deviation from the instructions** was present in several comments, either as a result of potential misunderstanding or ignoring parts of the instructions. For instance, annotating a sentence containing both positive and negative expressions as 'mixed' without specifying whether the strength of the sentiment mattered may ignore the important instruction to evaluate the strength of the contradicting sentiments.

> When I chose a sentiment label for each task I paid attention for key words in a sentence that gave me an idea if the word was a positive or a negative or if these words where not present I would label it as neutral or if both were present I would label them as mixed. (P106)

> I think there was an example in the instructions where "England beat France at the football" was supposed to be positive, but "England lost to France" was negative. I really don't care about football, England, or France, and these are factual statements not opinions. In the task there were "Norway beat Sweden at the ice hockey" and "Norway lost to Sweden" so I marked these as neutral. (P30)

> Sentences that had no keywords were neutral and sentences that had both negative and positive keywords were labeled as mixed and that was the easy part of the task. (P31)

**Inter-session changes.** Changes in the annotation experience between sessions were described as higher confidence in the second session, attributed to learning, or suspecting discrepancies between annotations across sessions, attributed to either learning or variability in one's mood.

> Doing the first part of the labelling, I understood what to do for the analysis but I second guessed myself on some of the sentences. However, this time around after reading the explanations before the labelling, I was more certain on the labels and could more confidently choose an answer them without going back and redoing them. (P11)

The way my mind read sentences could be different another time and be perceived as more positive/negative, etc. Saying that, some of my responses from last time could be the opposite to what I answered now just depending on my mood. (P52)

The second part I found easier as I reviewed the instruction list for a second time and had to make judgements on mostly the same set of questions. However some were more difficult as I know I selected some sentiments in Part 1 that in retrospect were incorrect due to reviewing the instructions for a second time which created conflicting feeling. (P26)

**Minimal group**

**Challenging aspects.** This theme encompasses the most challenging aspects of the annotation task, divided into three sub-themes.

**The volume of work** was mentioned as a challenge, particularly relating to increased effort to sustain focus over time.

Once you get towards the late-middle stage and the end, fatigue starts to set in and it becomes necessary to concentrate more on the question before it's answered. (P3)

The exercise did not last as long as I thought and I found it to be relatively easy. However it got very boring and monotonous very quickly as there were over 200 sentences in the first day and over 70 today (P56)

It was not very difficult to me but if I had to nitpick, I would say that it was a tiny bit difficult to stay focused on such a lengthy task. (P69)

**Challenging sentence types** included Sarcasm, Supplications & Requests, and Perspective-dependent sentences.

I found some sentences to be a little challenging to label because they contained a sense of neutrality but also had both positive and negative sentiments within them. For example the sentence: "The Lakers beat the Rangers 40-0" could be a positive sentiment for Laker's fans and a negative sentiment for Rangers fans and yet, at the same time, has a neutrality about it so I found those challenging to label. (P85)

There were some prompts that it was hard to tell if they were being sarcastic, and some prompts, like the more proverbial ones, the "let us bow our heads and pray..." where I found it hard to get a read on if it was mixed (i.e., hopeful because praying, yet sad about the things being prayed for) or if it was meant to be positive, or meant to be neutral. (P71)

**Interpretation** encompasses several challenges in judging sentiment, highlighting the factors that influence how a piece of text is interpreted. This includes who the speaker is, the context of the scenario described in the text, and who is reading the sentence.

It was mostly easy, some of the sentences were more difficult to rate because without more context it is hard to know what tone they would be said in and what the intentions are behind them. (P58)

Most of the labels i chose were easy to decide on, however there were some difficult ones where they were close to neutral/mixed and then there were quite a few that were positive/negative but depending on what point of view you look at the sentence from. (P59)

I think this task was really about perception. A lot of statements could have been seen differently from different views. for instance, there was one that stated about one team winning and one team losing. Your answer could have been all four depending on who is reading it. (P100)

**Spontaneous annotation criteria.** This theme consists of criteria or rules participants spontaneously adopted for their annotation task in the absence of clearer instructions. Two sub-themes were found.

**Mixed label for sentences with contradicting sentiment** was the primary criterion for most participants. This included any sentence with both positive and negative expressions of sentiment, regardless of the strength of these sentiments, and was thus often considered the easiest sentences in the annotation task.

> Regarding the mixed ones I chose, I felt these were easy because it was clear there was a positive and negative to the sentence. (P13)

> The easiest one I found was the mixed label. Any sentence with two clauses that contradicted each other, I used the mixed label. For example, "I really liked the movie (positive), but my wife hated it (negative)", I placed as mixed. (P4)

**Appraisal-based labelling of sentiment** was mentioned by a couple of participants, where the described event or situation was assessed for its positive or negative value, going beyond explicit expressions of sentiment.

> Some of them I found tricky to decide between neutral and positive, e.g., we've had breakfast together every Friday, because while it is factual, it has a positive experience within it. (P102)

> like for example in the question hope those rebuilding after the disaster find strength to keep going you could easily tell is positive because they are talking about rebuilding something so for me it's positive. (P44)

**Annotation approach.** This theme consists of different cognitive approaches to the annotation task, characterised by reliance on different aspects of the text samples when making annotation choices. Four sub-themes were identified.

**Word-focused approach**, in which sentences were scanned for sentiment-carrying words and other linguistic cues, such as contrastive connectives (e.g., 'but', 'however'), number of clauses, or whether the sentence was in the first person). Mechanistic and potentially shallower processing appears characteristic to this approach.

> I look at Positive or Negative words. "Won, Great, good, survived," those are positive. I cant remember other words. "Lost, hurt, pissed, jealous, disliked", those are negative. Usually both in one sentence is mixed. (P1)

> So I typically looked out for decriber words like "love, happy, smiled" for positive indication and the opposite for negative. The mixed ones usually, had a 'but' or 'however' which most of the time indicated mixed feelings. (P20)

**Holistic approach**, in which the overall emotional tone of the sentence was determined, often based on intuitive interpretation of the text, including emotional response to reading the text.

> When I was chooseing a sentiment label for each task, I first read each statement and then thought what my initial gut reaction was. (P32)

> The overall "mood" of the sentence was the primary driver of my response and for most sentences, it was immediately apparent. (P27)

> When I chose a sentimental label for a sentence, I paid attention to the tone. Tone really tells us a lot about the mood of what is being said. (P34)

**Word-focused and Holistic approach,** in which both word-focused screening of linguistic cues and more holistic interpretation of the overall emotional tone of the sentence were used.

If there was a 'but' or 'however' or 'though', it suggested that the sentiment would be mixed. The ones that sounded sarcastic I tended to opt for the opposite of the obvious answer, as I was reading it with the suggested sarcasm/irony in mind. (P102)

I had to think about the tone that I thought the sentence was being written in, and how many parts to the sentence there was. (P55)

My process for choosing a sentiment label mostly consisted of getting an overall emotion of what that sentence conveys. Usually greater negative or positive emotion would be conveyed if the sentence was speaking from first person. Sentences which seemed sarcastic, looked at the negative in things, appeared to be moaning, or had more negative than positive aspects, would be labeled as negative. […] I felt that for neutral sentences they were mostly straightforward and usually were easy to identify from a lack of first person. (P75)

**Empathic, role-taking approach,** in which participants imagine themselves as the speaker of the sentence and evaluate the sentiment they would experience when producing the sentence.

i found it easy, all i had to do was imagine myself in those scenarios and how i would feel in that situation. (P16)

I literally put myself in the position and thought about how I would see it – would I view it as positive or negative, etc.? If I felt that something was both positive and negative, I would choose 'mixed' and If I didn't know, I would go for 'neutral'. (P7)

**Clarity and brevity of instructions.** The instructions were generally deemed clear, however, some participants voiced the need for more examples, particularly for the 'neutral' sentiment.

The task generally was fun for me, the instructions was very clear and the task was put in a way that it was easy to complete and understand, although an example of a neutral sentiment wasn't given like the others (positive, negative and mixed), I was still able to find my way around it case of how clear the instructions were at the beginning. (P41)

I think one thing that could be improved, is to include an example of a neutral sentence as well. […] It might also be helpful to have the example sentences written above the sentences that need to be evaluated, so that participants can quickly reference back to them while doing the task. (P2)

As far as improving the task I would have more examples and more training before actually doing the task. (P86)

### Discussion of thematic analysis findings

Here, we reflect on the key findings from the reflexive thematic analysis [67,68], positioning them in relation to existing theories and literature (for a summary of the themes from both groups, see Fig 9). The focus of this discussion is on the identified cognitive approaches to annotation, spontaneous annotation criteria, and patterns of engagement with the instructions. These are discussed with the aim to contextualise the findings with prior literature, and to explore their broader implications for understanding cognitive and experiential aspects of annotation work.

**Cognitive approaches to annotation.** The three main cognitive approaches to sentiment annotation across instruction groups were word-focused, holistic, and emphatic role-taking. A combination of the first two approaches was also common, and these approaches may also change based on the type of sentence in question.

These approaches could be understood with different dual-processing accounts of higher cognition [79], which generally distinguish between two types of processing: Type 1 processing (often faster, intuitive, automatic, requires less conscious attention) and Type 2 processing (often slower, reflective, analytical, requires more conscious attention).

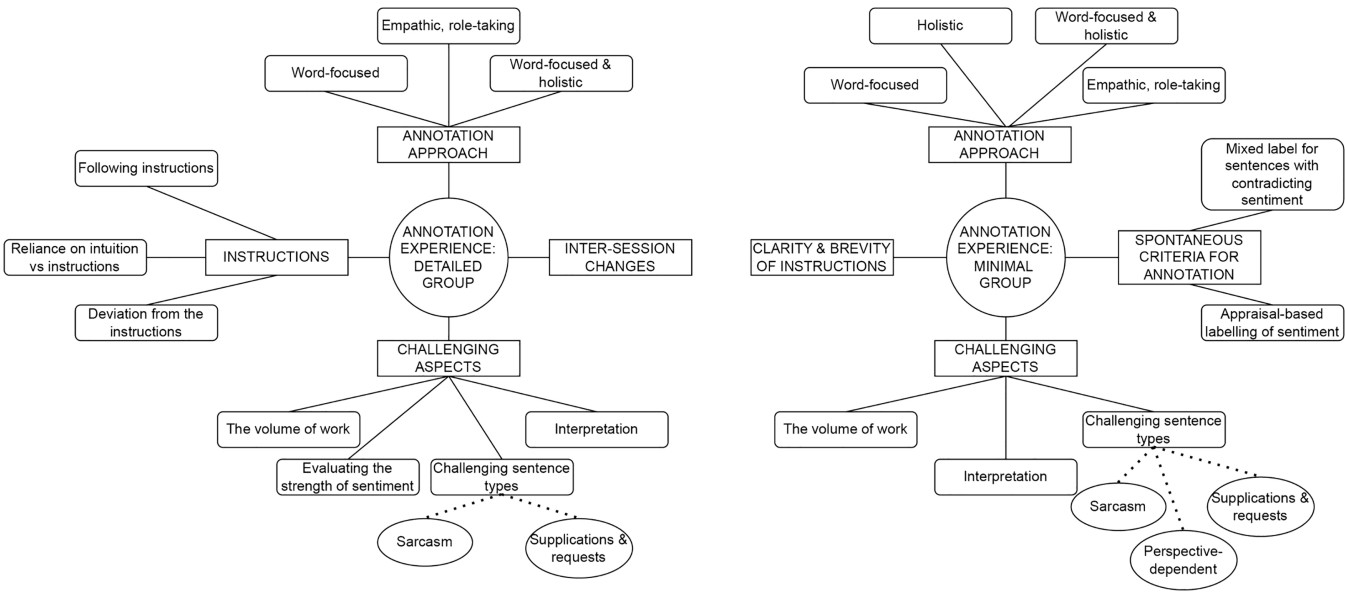

**Fig 9. Thematic map of annotation experiences.**

With these general characteristics, we discuss how the identified annotation approaches map onto the processing types of dual-process theories. The word-focused approach appears to involve mostly Type 1 processing, where surface-level linguistic cues are learnt (either via conditioning, or an initial, more effortful rule-seeking) and then used as a shortcut in decision making, reducing the level of semantic engagement with the full sentence. This is also akin to Kahneman's [80] substitution, where a difficult judgment is replaced, often unknowingly, with an easier judgement (e.g., estimating the probability of a rare event might be unknowingly replaced by judging how easily examples of the event come to mind). However, the fact that participants described this approach in such detail suggests that there was a conscious, non-automatic component to this approach, perhaps the initial rule-discovery process, where particular linguistic cues and patterns were used to aid quick decision making. This would position the word-focused approach somewhere between the Type 1 and Type 2 processing.

The holistic approach is best characterised as Type 1 processing, with reliance on intuitive linguistic understanding. Yet, this approach appears more semantically engaged with the whole sentence, compared to the word-focused approach. It may represent a more natural reading, where speed of decision making is not as central. This type of reading process may also be difficult to describe retrospectively, because much of it happens automatically.

The empathic, role taking approach, by contrast, corresponds more clearly with the Type 2 processing. This approach also aligns with concepts of cognitive empathy or mentalizing in theories of empathy [81–83], where understanding another's emotional state requires deliberate perspective-taking and mental simulation. This type of empathy is different from emotional empathy or experience sharing, which are considered more automatic processes. Participants describing the empathic role-taking approach often reflected on their annotation decisions in explicitly cognitive terms, suggesting engagement with controlled processes. In contrast, the more intuitive 'sensing the tone' described in the holistic approach may bear greater resemblance to emotional empathy, involving immediate, affective understanding.

Considering these cognitive approaches in terms of dual-processing accounts, it is worth noting that the distinction of Type 1 and Type 2 processing is also connected to more specific investigations and debates regarding automatic and controlled processes of reading [84,85], and investigations into the nature and timing of emotion processing when

reading words and sentences [86,87]. For instance, electroencephalogram (EEG) recordings can reveal distinct patterns of brain activity linked to emotion processing. Some of these patterns reflect early, automatic attention to emotional cues, while others are associated with more reflective, evaluative processing of emotional meaning [88,89]. For the purposes of understanding annotator behaviour, these lines of research should be informative for further theory development and empirical investigations.

**Spontaneous annotation criteria.** The approaches adopted by the Minimal group in the absence of clearer instructions consisted of 'Mixed label for sentences with contradicting sentiment' and 'Appraisal-based labelling of sentiment'. The former seemed to be particularly prevalent, also found in the comments from the Detailed group (see the sub-themes Reliance on intuition vs instructions and Deviation from the instructions). As such, there appears to be a strong intuitive tendency to interpret sentences with contradicting sentiment as 'mixed'. However, this tendency also simplifies the task significantly, reducing the required cognitive effort in sentiment judgements. Therefore, this tendency can also be understood as substitution [80,90], where participants replace interpreting the main message of the sentence (including evaluation of the strength of contradicting sentiments) with identifying whether a sentence contains both positive and negative sentiments. Arguably, detecting the dominant sentiment in a message is often more informative and required for many purposes, although it may not always be necessary. It appears that many participants in our study deemed it unnecessary for completing the annotation task. This relates to another common finding in the literature, namely, that participants modify their processing styles and strategies based on task demands [91–94]. Overall, the spontaneous annotation criterion to label sentences with contradicting sentiment as 'mixed' can be understood in terms of the resource-rational framework [93], which conceptualizes human cognition as the optimal use of limited computational resources.

The second spontaneous criterion, 'Appraisal-based labelling of sentiment', was described by a few participants in the Minimal group. This approach is characterised by an interpretation of text that goes beyond direct sentiment expression, such as evaluating whether the described situation would typically elicit positive or negative sentiment. This type of language processing is directly related to appraisal theories of emotion [95], according to which appraisal, i.e., the evaluation of the environment relative to one's well-being, is central to differentiating between emotions. Several appraisal criteria are proposed in these theories, such as an event's congruence and relevance to one's goals, and the level of agency, coping potential or control over the event. These types of evaluations impact the intensity and type of emotions that are elicited in a given situation. Understanding textual sentiment or emotion detection from the perspective of appraisal theories suggests that human readers should be capable of interpreting emotional content from text even without direct reference to emotions.

Some work in the area has been conducted – for instance, Troiano and colleagues [96] show that human annotators can extract appraisals and emotions from text, when instructed to do so. When and under what conditions human readers tend to do this spontaneously may be elucidated by literature reporting a processing advantage for emotion words with direct reference to emotion (e.g., sad, happy), and emotion-laden words referring to concepts associated with particular emotions (e.g., wedding, funeral). Compared to neutral words, emotional content is processed faster, typically such that emotion-laden words show smaller or less reliable effects than emotion words [87,97,98]. As reading emotion-laden words is intimately related to appraisal-based reading, these findings suggest that sentiment and emotion cues, whether direct or inferable (i.e., emotion words vs emotion-laden words), tend to be processed with relative ease, perhaps indicating the relevance of such information. As such, appraisal-based processing may emerge naturally in annotation tasks, especially when instructions are minimal and readers rely on their intuitive understanding of text.

**Instructions** Finally, we briefly discuss the two important sub-themes regarding instructions in the Detailed group, namely, 'Reliance on intuition vs instructions' and 'Deviation from the instructions'.

In the former, the participants expressed tension between the instructions and their intuitive approach to annotation. This can be understood as misalignment of schemas [99], where the instructional content differs from the annotators' existing interpretive schemas, that is, their prior knowledge, expectations, or intuitive strategies for interpreting sentiment. This misalignment can lead to confusion, hesitation, or inconsistent application of the guidelines [100,101].

The 'Deviation from the instructions' sub-theme reflected how participants either overlooked or misinterpreted the instructions. This may also be understood in terms of cognitive simplification and participants' tendency to strive for resource-efficient processing, as discussed above in 'Spontaneous annotation criteria'. However, it is also plausible that overly detailed instructions induce cognitive overload, which occurs when the amount of information presented exceeds the annotator's working memory capacity [102]. This is particularly likely when annotators are required to apply complex or conflicting criteria during rapid judgment tasks [103]. Such cognitive strain can lead to decision fatigue, where sustained mental effort reduces cognitive resources over time, resulting in less consistent responses or reliance on shortcuts [104].

## Integration of quantitative and qualitative findings

In this section, quantitative and qualitative findings are brought together to provide a more comprehensive understanding of the results as a whole. The integration focuses on how qualitative insights may help explain and contextualise the unexpected quantitative results. The integration of the findings is summarised in Table 7. This table also serves as a summary of the quantitative findings, including those that were not integrated with relevant qualitative findings. (For a summary of the qualitative findings, see Fig 9).

Taken together, the integration of quantitative and qualitative findings suggests the following explanation for the unexpected quantitative results. The higher inter-annotator agreement observed in the Minimal group may reflect a shared, intuitive annotation strategy – particularly for sentences with conflicting or appraisal-based sentiment. In contrast, agreement between annotators in the Detailed group was likely reduced by a subset of participants who found certain aspects of the instructions counter-intuitive, particularly regarding conflicting sentiments, resulting in misunderstandings or divergence from the guidelines. For intra-annotator agreement across sessions, the lower annotation consistency found in the Detailed group may reflect learning effects between sessions.

Underlying cognitive mechanisms that may account for these patterns include cognitive overload in the Detailed group, leading to inconsistent application of the instructions; schema misalignment, where the instructions conflicted with annotators' expectations; and cognitive optimisation strategies, such as substitution, used by annotators in both groups to reduce task demands.

To further explore the interpretation of the integrated findings, we conducted two additional quantitative analyses.

## Additional analysis 1

The integrated results suggest that some Detailed group participants deviated from the instructions, particularly when annotating contradicting sentiments. This would be particularly common for Two opinion holders and Two targets sentence types, for which the inter-annotator agreement in the Detailed group was one of the lowest. Inspecting the label choice distribution for these sentences further supports this interpretation. Namely, the Detailed group was clearly divided regarding sentences that required a 'positive' or a 'negative' label according to the instructions. On average, these sentences were annotated as 'mixed' by approximately 50% of the participants and 'positive' or 'negative', aligning with the instructions, by approximately 40% of participants. By contrast, the Minimal group consistently agreed on the 'mixed' label for these sentences, which was chosen by over 90% of the participants. As such, this quantitative inspection of the label choices supports the idea that a significant proportion of Detailed group participants followed their intuition rather than the instructions when annotating contradicting sentiments. Furthermore, this was the case even amongst the highest-performing participants (n = 28, those who scored at least 70% correct against the initial ground truth), as the 'mixed' label was still chosen by approximately half of these participants when the instructions required a 'positive' or a 'negative' label. See Figs 10 and 11 for visualisations.

## Additional analysis 2

The idea of learning across sessions attributing to the lower intra-rater agreement in the Detailed group was further investigated with dependent samples t-tests, comparing the mean proportions of correct responses against the initial ground

**Table 7. Integration of Quantitative and Qualitative findings.**

| Hypothesis/ Research question | Quantitative findings | Qualitative findings | Integrated findings |
|---|---|---|---|
| H1: Detailed inter-annotator agreement > Minimal inter-annotator agreement | Reliable difference between groups in the OPPOSITE direction: Krippendorff's alpha 0.59 (Minimal) > Krippendorff's alpha 0.50 (Detailed) | Spontaneous criteria for annotation: Mixed label for sentences with contradicting sentiment (Minimal); Challenging aspects: Evaluating the strength of sentiment (Detailed); Deviation from instructions (Detailed); Reliance on intuition vs instructions (Detailed) | The Minimal group's higher inter-rater agreement likely reflects a shared intuitive approach: labelling contradictory sentiments as 'mixed'. In contrast, some Detailed group participants experienced tension between their intuition and the instructions, with difficulties evaluating sentiment strength as directed. This led some to deviate from the instructions, reducing inter-rater agreement within the group. |
| H1: Detailed intra-rater agreement > Minimal intra-annotator agreement | Difference in the OPPO-SITE direction approaching statistical significance: Percent agreement 0.80 (Minimal) > Percent agreement 0.77 (Detailed) | Inter-session changes (Detailed) group; Deviation from instructions (Detailed); Reliance on intuition vs instructions (Detailed) | The Detailed group's lower intra-rater agreement may be due to learning between sessions, with less misunderstandings about the instructions or stricter adherence to them in the second session. |
| RQ1: Challenging aspects of annotation | Insufficient context (both groups mostly agree); Sarcasm (Minimal mostly agree, Detailed divided); Checking instructions and Instructions against intuition (Minimal mostly disagree, Detailed divided) | Challenging aspects: Sarcasm (both groups); Deviation from instructions (Detailed); Reliance on intuition vs instructions (Detailed) | Sarcasm was widely seen as challenging, especially without comprehensive instructions. The divided nature of responses to checking instructions and intuitiveness of the instructions suggests two subgroups within the Detailed group: those who largely followed the instructions, and those who did so only partially. |
| RQ2: Challenging sentence types | Supplications & requests (lowest Krippendorff's alpha in both groups); Appraisal, Two opinion holders & Two targets (largest difference in Krippendorff's alpha between groups: Minimal > Detailed) | Challenging aspects: Supplications & requests (both groups). Challenging aspects: Evaluating the strength of sentiment (Detailed); Deviation from instructions (Detailed); Reliance on intuition vs instructions (Detailed); Inter-session changes (Detailed); Spontaneous annotation criteria: Appraisal (Minimal) | Supplications & Requests were the most difficult sentences regardless of instructions. The two sentence types requiring sentiment strength evaluation showed much lower agreement in the Detailed group, likely as these judgments seemed difficult and counter-intuitive, leading some to deviate from the instructions. The Minimal group adopted appraisal-based labelling spontaneously, achieving higher agreement — possibly due to reliance on intuition, whereas the Detailed group may have second-guessed even intuitive judgments due to the extensive and partly counter-intuitive instructions. |
| RQ3: Diversity of annotation reasons | No difference between groups: mean entorpy H 0.90 (Detailed) and 0.89 (Minimal) | — | — |

truth across sessions, in both Detailed and Minimal groups. Neither group showed a statistically significant difference in correct responses across sessions (Detailed: $t(52) = 1.43$, $p = .16$, *Cohen's dz* = .20; Minimal: $t(52) = 0.92$, $p = .36$, *Cohen's dz* = .13). However, the Detailed group exhibited a trend toward improvement, with a p-value of 0.16 and slight increases in both mean (from 0.70 to 0.72) and median (from 0.71 to 0.74) scores. In contrast, the Minimal group showed no notable change, with nearly identical means (from 0.66 to 0.67) and medians (0.67 in both sessions), and a higher p-value of 0.36. The effect size estimate for the Detailed group was also larger compared to the Minimal group (*Cohen's dz* = 0.20 vs 0.13, respectively). The greater variability observed in the Detailed group ($SD = 0.11$ in both sessions, vs. 0.07 in both sessions in the Minimal group) may have contributed to reduced statistical sensitivity in detecting this trend. As such, these results provide tentative support to the integrated findings regarding the Detailed group's learning between sessions as an explanation for the lower intra-rater agreement.

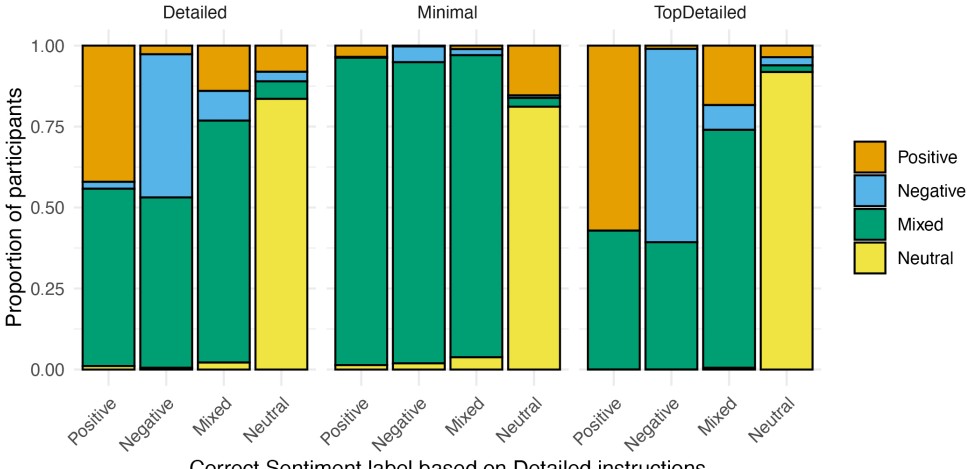

**Fig 10. Sentiment Distribution for sentence type Two opinion holders.** Detailed = full Detailed group (n = 53); Minimal = full Minimal group (n = 53); Top-Detailed = the best performing participants in the Detailed group (n = 28).

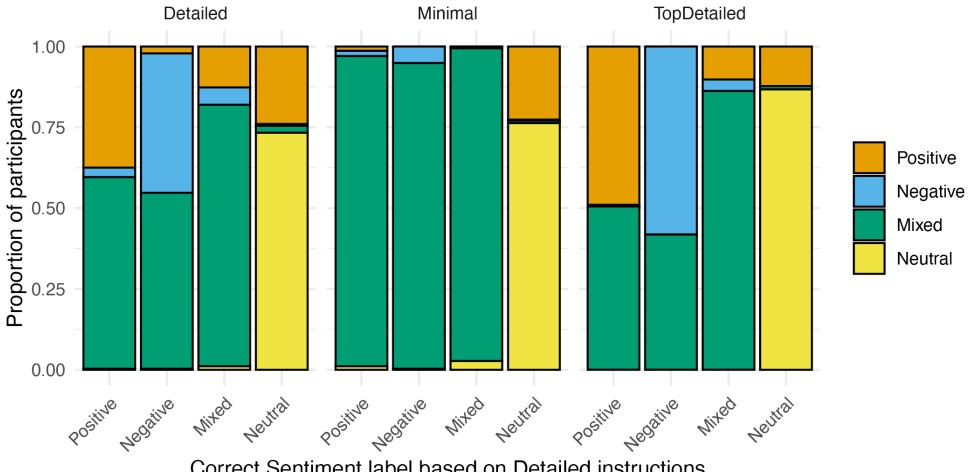

**Fig 11. Sentiment Distribution for sentence type Two targets.** Detailed = full Detailed group (n = 53); Minimal = full Minimal group (n = 53); Top-Detailed = the best performing participants in the Detailed group (n = 28).

## Discussion

In this section, we summarise our key findings and reflect on broader questions regarding annotation and its significance in automated text analysis. We also consider limitations of the study, outline future directions, and offer practical recommendations for conducting annotation projects, informed by our findings and relevant literature.

### Summary of key findings

We examine the main findings across three measures: inter- and intra-annotator agreement, which were the primary outcomes of interest, and proportion correct against the initial ground truth, which was used as an indicator of instruction adherence. While detailed instructions resulted in an overall higher proportion of correctly annotated items against the

ground truth (indicating instruction adherence), the overall agreement was lower in the Detailed group, both between and within annotators. By integrating the quantitative and qualitative findings, we show likely causes for this: 1) that some Detailed group participants found the instructions partly un-intuitive and thus deviated from the instructions, 2) the Minimal group's spontaneous approach to annotating some of the sentence types aligned with the detailed instructions, 3) due to the considerable length of the detailed instructions, the Detailed group second-guessed their labelling choices, even for some intuitive judgments, and 4) the Detailed group may have learnt more between sessions, thus changing their responses across sessions.

Regardless of instructions, the most challenging aspect of the task was insufficient context when reading the sentences, in line with findings from previous studies examining annotator experience [25,26]. With minimal instructions, annotators also found sarcasm and not knowing how much to interpret the text particularly challenging, in agreement with previous literature [26].

The most difficult sentence types, regardless of instructions received, were Perspective, Rhetorical, Sarcasm, and especially Supplications and Requests. Annotators tended to agree on these sentences less, and label these incorrectly against the initial ground truth. However, the detailed instructions also appeared helpful here: although the benefit was not seen consistently across all three measures (inter-rater and intra-rater reliabilities, and proportion correct), the Detailed group tended to show slightly better performance on most of these measures for Perspective and Rhetorical sentence types.

## What should be captured by human annotations?

Our guiding principle in these investigations has been that clear definition of what needs to be annotated reduces unnecessary noise in annotations (i.e., different interpretations of how the task should be done). Importantly, several researchers have advocated for minimal instructions in annotation tasks to better capture *natural* human judgement [46,105]. While this approach may be valuable for certain purposes, we argue that it presents an overly optimistic and simplistic view of annotator behaviour. Firstly, it is well established that humans tend to minimize cognitive effort [106–108]. In the absence of detailed guidance, annotators may resort to shortcut strategies, completing tasks quickly through shallow processing rather than deep linguistic analysis. Secondly, in many applications, the desired ground truth for AI systems is not superficial interpretation, but rather the best performance humans are capable of — careful, attentive language processing. Providing clear and detailed instructions supports this goal by helping to define what constitutes a meaningful linguistic judgment, such as sentiment labelling, and ensuring that annotations reflect engaged, deliberate decision-making rather than rushed judgment. Awareness of annotator behaviour, such as human tendency to avoid cognitive effort (see 'Discussion of thematic analysis findings'), encourages different interpretation of inter-rater reliabilities and aggregate labels: the majority label might still reflect heuristics and shortcuts in human language processing, rather than the ground truth intended for the purposes of a given dataset.

This question also relates to broader issues in sentiment analysis research, such as the elusive definition of sentiment. Sentiment is conceptualised in a variety of ways (or not at all) in several sentiment analysis studies [1]. Importantly, it is operationalised – explicitly or implicitly – through the procedures used to create different language resources. When this is done without careful consideration, the validity of the measured construct is compromised, leading to datasets that may reflect annotator biases or task-specific strategies rather than a clearly defined notion of sentiment. Constructing detailed annotation instructions essentially defines what sentiment detection is (or should be) for a particular dataset. Importantly, however, this definition varies based on what sentiment analysis is used for – some use cases may require more conservative detection of sentiment, whereas more interpretative approach may be required for others. For instance, evaluating user satisfaction in staff surveys or customer feedback may require a conservative approach, where ambiguous or context-dependent expressions are classified as neutral rather than risking over-interpretation. By contrast, analysing opinion in literary studies [109] or political discourse may require a more nuanced approach that accounts for

subtle linguistic cues and inferred meaning. Overlooking the importance of clearly defining sentiment and specifying how it should be captured for a particular use case often reflects outdated assumptions about sentiment analysis (and more broadly, natural language processing), where a single, universal ground truth is presumed to exist. This belief is often paired with the assumption that annotators will intuitively arrive at the 'correct' interpretation, even in the absence of explicit guidance. However, texts are frequently open to multiple valid interpretations, influenced by factors such as the annotator's background, goals of the task, and clarity of instructions. Expecting consistent, high-quality annotations without clearly communicating what the task requires places unrealistic demands on annotators and undermines the reliability of the resulting data. This is especially so for typical annotation projects where the ground truth is not known beforehand, but emerges as aggregate annotations. As shown in this study, high annotator agreement does not necessarily correspond to the intended ground truth, but may reflect cognitive heuristics that go against how sentiment was supposed to be captured in a particular dataset. As such, annotation instructions must not only be detailed but also effectively motivate annotators to adhere to them, especially when they conflict with personal intuition.

## Who should annotate the data?

The annotators in our study were crowdworkers, due to the requirement of relatively large sample sizes to obtain sufficient statistical power for the study, and the need for varied demographics. Although crowdworker annotations are often deemed of appropriate quality [110–112], it is nonetheless important to consider the circumstances under which crowdworkers perform the tasks. Many crowdworkers complete numerous tasks per day and rely on the income generated through crowdworker platforms [113,114]. Consequently, annotation becomes a balancing act where workers must complete tasks quickly enough to make the work financially viable, yet maintain a level of quality that ensures their submissions are accepted and they retain access to future tasks. On the other hand, expert annotators or hired in-house annotators are typically costly [112,115] and thus result in a considerably smaller sample of annotators. Deciding for what purpose the annotation project is conducted is therefore crucial, with some requirements of the project favouring crowdworkers over experts and vice versa. Crowdworkers are more suitable for projects where a large or a representative sample of annotators is required, where the annotation task can be completed without supervision and in a low number of sessions, and where annotations should be based on heuristics or intuitive approaches. By contrast, expert annotators or hired in-house annotators are more suitable for projects where the subject matter is complex and requires specialist knowledge [116,117], where extensive and/or iterative training is required, where the instructions are likely against intuitive annotation approaches, and where annotators with higher motivation or professional commitment are required. Additionally, the longitudinal nature of annotation work with hired annotators allows for quality control in the form of annotator accountability, continuous monitoring of the annotations, and calibration exercises or discussions regarding disagreements.

## Implications for sentiment analysis systems

Finally, our findings have practical implications for the development of machine learning systems trained on human annotations. First, the observation that detailed instructions do not consistently increase agreement highlights the need for caution in assuming that instruction-based standardization reduces label noise. In fact, instructions may introduce new inconsistencies if they conflict with annotators' intuitive judgments. Second, some observed disagreement or deviations from intended ground truth may stem from annotators relying on cognitive shortcuts or heuristics, especially under cognitive load (which is often the case when crowdworkers try to complete the tasks as quickly as possible). While such responses may be undesirable when the goal is to elicit attentive, fully engaged annotation, they nevertheless reflect a particular mode of real-world language processing – namely, how individuals might interpret sentiment when attention is limited or contextual nuance is overlooked. In this sense, heuristics-based annotations could be useful for modelling 'fast' or intuitive judgments, as might occur in social media interactions or real-time applications such as chat moderation. Capturing both effortful and heuristic modes of human judgment may therefore support the development of systems tailored

to different types of language use, depending on whether the goal is to model careful reflection or fast, intuitive interpretation. We therefore highlight the importance of critically assessing what form of ground truth is most suitable for the specific goals of a given application. With this in mind, our findings also support emerging practices in machine learning that move beyond single hard labels, and instead preserve the distributions of annotator responses via soft labels. Such approaches allow systems to account for more of the psychological reality of sentiment judgment, and offer end-users flexibility for deciding if and how disagreement is handled for different use cases.

## Limitations

Due to the already time-consuming study design (with two over an hour-long sessions, seven days apart), it wasn't feasible to screen participants for performance beyond the attention checks. This in part resulted in a sample with more variability in task engagement than desired. However, we also show that even participants with high proportion of matches to the initial ground truth still resort to their intuition over the instructions for particular items (Two opinion holders and Two targets), thus revealing an important consideration for future annotation projects.

Additionally, our choice to prioritize experimental control and ethical data sharing by using crowdworker-generated sentences may limit ecological validity of the dataset. Future research should therefore examine whether the observed patterns hold in more naturalistic language contexts. Furthermore, our work was based on crowdworkers and native English speakers, which therefore limits the generalizability of the findings to other annotator populations, such as expert annotators, or annotators from different cultural or linguistic backgrounds (see 'Future directions').

Finally, we did not attempt to isolate instruction-related ambiguity from other forms of ambiguity in sentiment judgments, due to the practical and conceptual challenge of creating a truly 'unambiguous' baseline for something as inherently subjective as sentiment. Even in our 'Explicit' condition (e.g., 'He is completely terrified of any kind of insect'), it is not necessarily clear whose sentiment should be assessed: the speaker of the sentence, or any opinion-holder (i.e., 'he'). Instructions are needed to clarify this ambiguity. However, future work may endeavour to constructsuch a baseline through careful norming and validation, and thus help isolate different types or sources of ambiguity in sentiment judgments.

## Future directions

In our discussion of thematic analysis findings, we connect annotators' descriptions of the annotation process with psychological theories and research of decision-making and emotion processing. This type of bridging is important for characterizing and understanding annotator behaviour in relation to broader frameworks of human cognition. Further, it opens several new avenues for investigation. For instance, if particular annotation approaches (e.g., word-focused or empathic, role-taking) can be systematically identified, future research may explore how to develop more targeted task designs, including instructions that either encourage or discourage particular cognitive approaches, depending on the goals of the annotation project. Ultimately, bridging psychological research and annotator behaviour holds the potential to enhance both the theoretical understanding of human language processing and the practical development of more robust language resources.

With the current developments and interest in generative AI and/or large language models (LLMs), some work is already looking at how generative AI models may be utilised in data annotation tasks [118,119], and in sentiment annotation [120,121]. These investigations often compare agreement measures between human annotators and LLMs, and generally conclude that the combination of human and LLM annotations may be beneficial. However, this enthusiasm appears premature, considering the reports and observations regarding hallucinations [122] and drastic changes in LLM performance over time [123]. Many investigations comparing human annotators to LLM performance also appear to equate high level of agreement with 'correctness' of a label, which, as we have shown, is not necessarily the case.

Another important direction for future work could involve exploring how annotators' cultural and linguistic backgrounds shape their interpretations of sentiment. Sentiment is often culturally situated, and the use of sarcasm, irony, or emotional intensity can vary significantly across linguistic communities [124,125]. For example, expressions of negative emotion may

be more subdued in collectivist cultures compared to individualist ones, while forms of indirect speech such as sarcasm may be more easily recognised or socially accepted in certain linguistic or regional contexts [124,126]. Future research could therefore investigate how culturally grounded expectations and values influence annotation outcomes. This could involve cross-linguistic annotation tasks or demographically diverse annotator pools and may help uncover sources of variability not attributable to instructions or fidelity to instructions alone. Such work would deepen understanding of both inter-annotator disagreement and the cultural embeddedness of sentiment.

### Recommendations

Our study highlights important aspects of annotator behaviour that should be considered in future annotation projects. Below, we list recommendations drawing from our findings and relevant literature, particularly in psychology.

- Identify particularly challenging sentence types or text samples based on previous literature, piloting, or annotation quality measures of the dataset (e.g., annotator agreement or fidelity to ground truth). These items should receive greater emphasis during training. For example, our study shows that sentences requiring evaluation of sentiment strength are particularly challenging: for projects where identifying the dominant sentiment is important, more training should focus on this particular sub-task.

- Identify areas of the annotation task where cognitive load is likely to be high. For example, where multiple decision rules need to be applied simultaneously, or where criteria appear to conflict. To reduce working memory demands, break instructions into smaller sections or hierarchies, and clearly distinguish core rules from exceptions. This should improve adherence to annotation instructions.

- Use pilot testing, annotator feedback, or think-aloud protocols to identify points of confusion or misalignment between the instructions and annotators' intuitive strategies. Pilot studies can help surface unclear phrasing or unexpected interpretations and are increasingly recognised as essential in annotation design [127]. Think-aloud protocols are also effective for capturing real-time cognitive responses and identifying where instructions conflict with annotators' expectations [128]. Revise wording or structure to improve clarity and conceptual fit, and use representative examples to illustrate more ambiguous or cognitively demanding cases. Allocate more training for annotating challenging text samples according to the instructions, to improve annotator agreement and adherence to instructions.

- Spread training across multiple sessions, in line with spacing effect – a learning principle showing that information is better retained when exposure is spaced over time [129,130]. This aligns with our observation that some participants in the Detailed group found the annotation task easier in the second session, likely due to consolidation of the instructions. Spaced learning procedures should improve annotation quality and wellbeing of annotators on large-scale projects.

- Where feasible, provide feedback on practice items. Feedback enhances learning by enabling error correction and reinforcing correct strategies, particularly in procedural and judgment-based tasks [131,132]. This should be particularly useful for learning instructions that are more demanding or less intuitive, which should improve annotation quality.

- Enhance annotator motivation, as it is critical for maintaining attention and task engagement [133]. Annotators may be motivated by fun, profit or altruism, and different crowdworker platforms are typically characterised by particular sources of motivation [134]. Consider strategies such as ensuring fair monetary compensation (especially for crowdworkers), introducing gamification elements to the task, or fostering a sense of accountability, such as identifying annotators and tracking reliability over time.

- Limit the volume of annotations per session to preserve focus. Prolonged task engagement leads to cognitive fatigue and reduced performance, especially for repetitive and attentionally demanding tasks [70,135]. Limiting the volume of items per session can prevent decline in annotation performance.

- Ask for feedback during and afterwards to identify potential issues (including unexpected approaches to and interpretations of the task), and to continue accumulating knowledge regarding annotator experience and behaviour during annotation tasks. This builds rapport between annotators and the project leads, which in turn should improve annotation quality, the project outcomes, and annotator wellbeing.

- Where disagreement between annotators likely reflects differing views and interpretation rather than error, consider preserving variation in the form of soft labels rather than collapsing responses into a single label. This enables systems to learn from uncertainty and may improve calibration and generalisation [40,41].

- Where possible, use disagreement-aware modelling approaches that treat disagreement as informative. For example, models can weight training instances based on agreement level, account for annotator-level differences, or model response distributions directly rather than assume a fixed ground truth [37,41]. This should improve the overall project outcomes.

## Conclusion

In this mixed-methods study, we investigated annotator experiences and the role of instructions in annotator agreement. We show that detailed instructions do not guarantee more consistent annotation across or within participants. The utility of detailed instructions appears to depend on participants' task engagement and the extent to which the instructions align with annotators' intuitive approaches. We find a strong tendency for participants to simplify the annotation task regardless of instructions (such as overlooking contradicting sentiments within a sentence), but also cases where the instructions may lead to higher inter-annotator agreement (particularly for sentences involving shifts in perspective, or rhetorical phrasing).

Our discussion of annotation as a cognitively effortful process, along with strategies for increasing task engagement and adherence to instructions offer practical insights for future dataset development. More broadly, these findings highlight the importance of recognising the interpretive strategies and cognitive shortcuts that annotators may employ, and call for deeper consideration of how such behaviours impact current and future modelling efforts and their integration into computational systems. As sentiment analysis and other natural language processing applications become increasingly integrated into organisational, social, and regulatory contexts, building robust and inclusive systems will require not only technical advances, but a deeper understanding of the human judgments that underpin them.

## Supporting information

**S1 File. Demographic questionnaire.**
(PDF)

**S2 File. Detailed instructions.**
(PDF)

**S3 File. Sentiment judgment questionnaire.**
(PDF)

## Author contributions

**Conceptualization:** Laura E. M. Äyräväinen, Joanne Hinds, Brittany I. Davidson.

**Data curation:** Laura E. M. Äyräväinen.

**Formal analysis:** Laura E. M. Äyräväinen.

**Funding acquisition:** Joanne Hinds, Brittany I. Davidson.

**Investigation:** Laura E. M. Äyräväinen.

**Methodology:** Laura E. M. Äyräväinen.

**Project administration:** Laura E. M. Äyräväinen.

**Resources:** Laura E. M. Äyräväinen.

**Software:** Laura E. M. Äyräväinen.

**Validation:** Laura E. M. Äyräväinen, Joanne Hinds, Brittany I. Davidson.

**Visualization:** Laura E. M. Äyräväinen.

**Writing – original draft:** Laura E. M. Äyräväinen, Joanne Hinds, Brittany I. Davidson.

**Writing – review & editing:** Laura E. M. Äyräväinen, Joanne Hinds, Brittany I. Davidson.

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
