## [Decision Letter · Decision Letter 0]

2 Jul 2025

Dear Dr. Hinds,

Thank you for submitting your manuscript to PLOS ONE. After careful consideration, we feel that it has merit but does not fully meet PLOS ONE’s publication criteria as it currently stands. Therefore, we invite you to submit a revised version of the manuscript that addresses the points raised during the review process.

We look forward to receiving your revised manuscript.

Kind regards,

Hessameddin Ghanbar

Academic Editor

PLOS ONE

Journal Requirements:

Reviewers' comments:

Reviewer's Responses to Questions

**Comments to the Author**

1. Is the manuscript technically sound, and do the data support the conclusions?

Reviewer #1: Yes

Reviewer #2: Partly

2. Has the statistical analysis been performed appropriately and rigorously?

Reviewer #1: Yes

Reviewer #2: No

3. Have the authors made all data underlying the findings in their manuscript fully available?

Reviewer #1: Yes

Reviewer #2: Yes

4. Is the manuscript presented in an intelligible fashion and written in standard English?

Reviewer #1: Yes

Reviewer #2: Yes

Reviewer #1: This manuscript presents an important and well-structured study investigating the impact of instruction detail on the quality of sentiment annotation. The study design is rigorous and scientifically sound, with appropriate sample size and power analyses. The data robustly support the conclusions drawn by the authors.

The statistical analyses are appropriate and carefully conducted, including the use of Krippendorff’s alpha for inter-annotator agreement, which is well suited for nominal data. Both descriptive and inferential statistics are applied correctly.

The authors have made all relevant data and materials publicly available on the Open Science Framework, promoting transparency and reproducibility.

The manuscript is clearly written in standard English, with no major language or structural issues, facilitating reader comprehension.

Suggestions:

1. Future work could benefit from exploring annotator diversity across different linguistic and cultural backgrounds.

2. A more detailed discussion on the advantages of soft labeling for model development would strengthen the manuscript.

3. Including a brief reflection on the ethical considerations related to participant exclusion due to failed attention checks would enhance transparency.

Overall, this manuscript is a valuable contribution to the field, with novel insights and sound methodology. I recommend acceptance after minor revisions.

Reviewer #2: Dear Authors,

While the study is well-intentioned and thematically relevant, it suffers from critical weaknesses across nearly every major section of the paper. These limitations undermine both the theoretical contribution and the reliability of the empirical findings.

The abstract, while clearly structured, fails to convey the most unexpected and important findings of the study—namely, that detailed annotation instructions did not improve, and in fact appeared to reduce, inter- and intra-annotator agreement. This counterintuitive result, which contradicts the study’s own hypotheses, is central to the paper's impact and should be explicitly flagged in the abstract. Moreover, the methodological framing is vague. Key elements such as dataset size, the number of annotators, the nature of the statistical measures employed, and the form of qualitative analysis are all omitted, reducing the abstract's utility as an informative summary.

The introduction, while providing adequate background on the importance of sentiment annotation in NLP, suffers from conceptual underdevelopment. The authors introduce the idea of “interpretive ambiguity” but fail to define it rigorously or connect it to established theoretical frameworks from discourse analysis, pragmatics, or subjectivity studies. The discussion of sentiment is similarly reductive, conflating it with emotion, evaluation, and stance without engaging with the rich multidisciplinary literature that differentiates these concepts. Furthermore, the critique of existing datasets is primarily descriptive and lacks the depth needed to justify the construction of a new dataset. There is insufficient examination of how previous datasets have handled ambiguity or why existing annotation protocols are inadequate. This results in a justification for AmbiSent that appears more speculative than necessary.

The articulation of hypotheses and research questions is another area of concern. The authors posit that detailed instructions will increase both inter- and intra-annotator agreement, but they offer no compelling theoretical or empirical rationale to support this assumption. Moreover, the operational definitions of key terms such as “annotation quality,” “diversity of reasons,” and “challenge” are imprecise. The research questions are loosely formulated, overlapping in scope and lacking clear analytical boundaries. There is also no hierarchical organization among the hypotheses and exploratory questions, which complicates interpretation of results and weakens the study’s inferential structure.

The methodology section is detailed in parts but lacks essential clarity in others. Most notably, the instructional manipulation—comparing “minimal” versus “detailed” instructions—raises concerns about ecological validity. The level of detail provided in the “detailed” condition is extensive and arguably unrealistic for real-world annotation workflows, which typically involve minimal training to ensure scalability. There is no empirical evidence that annotators actually absorbed or adhered to the complex instruction set, nor is there any direct measure of cognitive load or comprehension. Additionally, while the authors emphasize their commitment to interpretive subjectivity by using soft labels, they paradoxically use native speaker judgments to define preliminary ground truths, a move that contradicts their own critique of fixed labeling. This contradiction undercuts the theoretical coherence of the study.

The construction of the AmbiSent dataset introduces further problems. The dataset consists entirely of artificially constructed sentences, which, while enabling control over sentence types, severely limits ecological validity. The sentences are not sampled from naturalistic corpora, such as Twitter, Reddit, or customer reviews, but are instead created by crowdworkers for the express purpose of being ambiguous. This makes it unclear whether the types of ambiguity captured are reflective of those typically encountered in real-world sentiment classification tasks. Moreover, the dataset lacks any baseline set of unambiguous items, making it difficult to isolate the effect of ambiguity from the effect of instructions. There is also no linguistic analysis of the dataset in terms of sentence complexity, length, syntactic structure, or lexical difficulty—all of which are known to influence annotation behavior.

The statistical analyses conducted are limited in scope. The authors rely primarily on Krippendorff’s alpha as their measure of agreement, without triangulating results with other relevant metrics such as Cohen’s kappa, Fleiss’ kappa, or Gwet’s AC1. Confidence intervals and effect sizes are not reported, and there is no mention of multiple comparison corrections, despite the number of tests conducted. The analysis does not account for annotator- or item-level variation through mixed-effects modeling, nor does it explore potentially confounding variables such as annotator demographics, prior experience, or annotation speed. These omissions weaken the robustness of the quantitative claims.

The qualitative strand of the study, based on thematic analysis of annotator feedback, is underdeveloped. The analysis was conducted by a single coder, with no evidence of reliability checks, coding framework development, or triangulation. The themes presented are not supported by participant quotations or tied back to theoretical constructs. There is no engagement with established methods in qualitative research, such as grounded theory or discourse-oriented coding, and no attempt to integrate qualitative insights meaningfully with the quantitative findings. Most notably, the authors miss the opportunity to use qualitative feedback to explain the unexpected result that minimal instructions outperformed detailed ones—a finding that could have significantly benefited from nuanced qualitative interpretation.

In the discussion and conclusion, the authors acknowledge the surprising outcome that more instructions led to less consistent annotation, but their explanation is superficial and largely speculative. There is no engagement with cognitive theories that might explain such findings, such as cognitive overload, decision fatigue, or schema misalignment. The implications for future annotation design are also vague. While the authors suggest that more attention should be paid to instruction crafting, they provide no practical guidelines or frameworks to support this claim. Similarly, there is little discussion of how their findings might inform automated methods for managing label noise or disagreement in machine learning systems.

Overall, while the manuscript tackles a compelling problem and demonstrates a commendable attempt at methodological triangulation, the study suffers from serious conceptual and methodological limitations. These include a lack of theoretical grounding, ecological validity issues in dataset design, incomplete statistical reporting, a superficial qualitative analysis, and insufficient integration between study components. These limitations collectively reduce the manuscript’s contribution to the field of annotation science and sentiment analysis.

Kind regards,

Reviewer

**Do you want your identity to be public for this peer review?** For information about this choice, including consent withdrawal, please see our Privacy Policy

Reviewer #1: No

Reviewer #2: No

---

## [Author Response · Author response to Decision Letter 1]

22 Aug 2025

Point by Point Response

Reviewer #1 Comments

Reviewer #1: This manuscript presents an important and well-structured study investigating the impact of instruction detail on the quality of sentiment annotation. The study design is rigorous and scientifically sound, with appropriate sample size and power analyses. The data robustly support the conclusions drawn by the authors.

The statistical analyses are appropriate and carefully conducted, including the use of Krippendorff’s alpha for inter-annotator agreement, which is well suited for nominal data. Both descriptive and inferential statistics are applied correctly.

The authors have made all relevant data and materials publicly available on the Open Science Framework, promoting transparency and reproducibility.

The manuscript is clearly written in standard English, with no major language or structural issues, facilitating reader comprehension.

Suggestions:

1. Future work could benefit from exploring annotator diversity across different linguistic and cultural backgrounds.

2. A more detailed discussion on the advantages of soft labeling for model development would strengthen the manuscript.

3. Including a brief reflection on the ethical considerations related to participant exclusion due to failed attention checks would enhance transparency.

Overall, this manuscript is a valuable contribution to the field, with novel insights and sound methodology. I recommend acceptance after minor revisions.

We’d like to thank you very much for your kind and thorough feedback both on the manuscript itself and the above summarised comments. We are pleased to hear you see substantial merit in the manuscript and found the methodology thorough and sound.

Our response to the main suggestions are as follows:

1. We have added this perspective to our discussion of future directions, as follows:

“Another important direction for future work could involve exploring how annotators’ cultural and linguistic backgrounds shape their interpretations of sentiment. Sentiment is often culturally situated, and the use of sarcasm, irony, or emotional intensity can vary significantly across linguistic communities [124,125]. For example, expressions of negative emotion may be more subdued in collectivist cultures compared to individualist ones, while forms of indirect speech such as sarcasm may be more easily recognised or socially accepted in certain linguistic or regional contexts [124,126]. Future research could therefore investigate how culturally grounded expectations and values influence annotation outcomes. This could involve cross-linguistic annotation tasks or demographically diverse annotator pools and may help uncover sources of variability not attributable to instructions or fidelity to instructions alone. Such work would deepen understanding of both inter-annotator disagreement and the cultural embeddedness of sentiment.”

2. We have elaborated on how soft labels may benefit model development as follows:

“Soft labels not only reflect the interpretive nature of sentiment in text more accurately, they can also be beneficial in system development. Rather than collapsing differing judgments into a single ground truth, soft labels preserve the distribution of annotator responses, enabling models to learn from uncertainty instead of treating it as noise [40,41]. This can lead to improved generalisation to unseen data [41]. For instance, training on soft labels tends to improve model robustness to adversarial attacks (i.e., subtle input manipulations that mislead models into incorrect predictions) [40,42]. These benefits arise because soft labels encourage models to maintain calibrated confidence levels and avoid overfitting to noisy or ambiguous examples [43,44]. They also help mitigate the effects of label noise, as soft targets reduce the influence of outlier annotations that might otherwise skew model training [45]. Moreover, by encoding degrees of annotator disagreement, soft labels promote a more epistemically transparent and psychologically realistic approach to classification, particularly in tasks involving subjectivity or ambiguity [46]. This not only improves model interpretability but aligns more closely with ethical practices in artificial intelligence (AI), where acknowledging uncertainty is often preferable to enforcing false certainty [47].”

3. Some discussion of ethical considerations related to participant exclusion due to failed attention checks was already part of the manuscript in section ‘Data collection and compensation’:

“Any participant who took longer than this and participants who were excluded from the study (see Exclusion of participants) received compensation at the same hourly rate, calculated pro rata based on the precise duration of their task involvement. Thus, we ensured a minimum compensation of £11.44/hour for all participants.”

However, to make this more explicit, we have added the following in the section ‘Exclusion of participants’:

“While exclusion due to failed attention checks or incomplete data was essential for maintaining data integrity, we recognize the ethical responsibility to ensure fair treatment of all participants. Accordingly, excluded participants were compensated pro rata for their time spent on the task, ensuring no participant was disadvantaged financially (see section Data collection and compensation).”

Below, we outline our response to each comment in the manuscript:

Reviewer #1 Comments from the manuscript (the PDF)

TItle: The current title is informative but rather long and dense. Consider shortening it or breaking it into two clearer conceptual parts to enhance readability.

Thank you for this comment. We have shortened the title slightly to increase readability. However, we still prioritized the level of information over brevity. We see it important for the title to reflect the breadth of the paper, which also helps maintain discoverability for researchers conducting literature reviews or searching for work on the key components of our paper (i.e., annotator experiences, impact of instructions, sentiment analysis, or mixed methods research). The title is now modified as:

“Disambiguating sentiment annotation: A mixed methods investigation of annotator experience and impact of instructions on annotator agreement”,

which hopefully increases the readability.

Abstract: The abstract is well-structured and effectively outlines the study’s rationale, methodology, key findings, and implications. It reflects a clear understanding of the problem and offers meaningful contributions.

The opening sentence could be more compelling. You may consider adding a concrete example or a brief statistic to emphasize the importance of human-annotated data in current NLP practices.

Thank you for this helpful comment! We have modified the opening sentence to convey how broadly annotated datasets are needed. However, along with other revisions, the word-limit for the abstract prevents us from adding much more information, such as statistic to highlight the importance of annotated datasets. The sentence now reads:

“Human-annotated datasets are central to the development and evaluation of sentiment analysis and other natural language processing systems.”

“...should learn from or align with” [abstract]:

slightly informal

Thank you, we changed the sentence to adopt a more formal tone:

“the quality of the data that is used to train and evaluate computational systems.”

“...label consistency” [abstract]:

When discussing label consistency, it would help to specify whether this refers to inter-annotator agreement, intra-annotator agreement, or both. Precision here strengthens the argument.

This is a helpful comment, we’ve clarified this section as follows:

“we investigate how annotation instructions influence annotator experience and both inter-annotator agreement (Krippendorff’s alpha) and intra-annotator agreement (percent agreement)”

“...simplifying the task” [abstract]:

may be too colloquial for a scientific abstract

We agree this could be expressed more formally, and thus made the following modification:

“participants often resorted to reductive annotation approaches.”

“Sentiment Analysis is a computational method for classifying subjective information, such as expressions of emotion, opinion and attitudes, from text and other content” [introduction]:

Consider revising to make it more engaging or problem-driven, for example by mentioning the prevalence of sentiment analysis in everyday technologies or its impact on decision-making systems. The reference to real-world applications in both sentence 2 and sentence 3 feels slightly repetitive. Consider merging or rephrasing for conciseness.

Thank you for this helpful comment! We have modified sentence 3 to reduce repetition and to incorporate your suggestion:

“Research interest in sentiment analysis is increasing rapidly [1-3], and the potential for real-world applications is met with enthusiasm across sectors, such as finance [4], education [5], and healthcare [6]. Given the widespread application of sentiment analysis in everyday technologies and its increasing impact on decision making, including high-stakes scenarios such as stock market prediction and healthcare applications, it is crucial to ensure that the outcomes from different sentiment analysis systems are valid and trustworthy.”

“Thus, human judgement forms the foundational standard for sentiment classification in computational systems.” [introduction]:

This is a key point, but it could be strengthened by linking it more explicitly to the issue of subjectivity.

Thank you for raising this point! To preserve the flow of the introduction, this addition can now be found two sentences later in the introduction, where other sources of annotation discrepancies are introduced:

“Thus, human judgement forms the foundational standard for sentiment classification in computational systems.

However, human annotators do not always agree on the appropriate sentiment label for a given text sample. In fact, discrepancies in sentiment labels are prevalent [9], and several datasets annotated for sentiment contain annotation errors [10,11]. Apart from the inherently subjective nature of sentiment interpretation, discrepancies in annotations can arise from several other sources, such as…”

“Given these issues of annotation discrepancies and the central role annotated datasets play in sentiment analysis system development and evaluation, a comprehensive understanding of how these datasets are produced is required.” [introduction]:

the sentence is long and may benefit from being split or simplified for clarity.

Thank you, we have simplified the sentence structure as follows:

“As sentiment analysis system development and evaluation rely on annotated datasets, yet, these datasets often suffer from annotation discrepancies, it is essential to understand how these datasets are produced.”

“The dataset includes soft labels that reflect annotation variability, along with detailed annotator demographics – characteristics that make it a valuable resource for advancing research in sentiment analysis and dataset construction. In this context, our study and the AmbiSent dataset provide a timely reminder of the often overlooked assumptions surrounding subjective annotation work, pariticularly how tasks are interpreted, what constitutes reliable agreement, and the nature of the ‘correct’ label.” [introduction]:

The sentence structure is quite complex.

Thank you, we have simplified the sentence structure as follows:

“The dataset includes soft labels that reflect annotation variability, and detailed annotator demographics, making it a valuable resource for research in sentiment analysis and dataset construction. Our study and the AmbiSent dataset highlight often-overlooked assumptions about subjective annotation work, particularly regarding how annotation tasks are interpreted, what constitutes reliable agreement, and how the ‘correct’ label is defined.”

Commented on the “Previous investigations of annotator behaviour and experience” section:

The motivation behind reviewing these specific studies could be more clearly stated at the beginning. Consider starting with a framing sentence like:

Understanding how annotators engage with the task is essential for improving annotation quality, yet remains underexplored in the literature.

This would help establish the section's relevance more strongly.

This is a helpful comment, and we have incorporated this suggestion as follows in the beginning of this section:

“Despite the essential role annotators play in shaping datasets, relatively little attention has been paid to how they engage with annotation tasks in practice. Understanding annotator behaviour and experience is crucial for improving annotation quality, such as annotator agreement, error rates, and fidelity to the intended ground truth. This would, by extension, enhance the reliability of systems trained on such data. Optimising the annotation process requires knowing how annotators might perform the task and what aspects of the task they find the most challenging.”

“A handful of studies” [Prev invest]:

The phrase “a handful of studies” appears twice in close proximity. While not incorrect, its repetition slightly weakens the formal tone

Agreed - we have since changed the wording to avoid repetition:

“Some studies have investigated this in different annotation tasks, […], and inter-rater agreement measures.

A small number ofSeveral studies have also investigated annotation as a cognitive process…”

“These eye-tracking measures were either used to gain insights into the annotation process, or leveraged in training a classifier.” [Prev invest]:

Somewhat vague

Thank you - we have since added more detail:

“These eye-tracking measures were either used to gain insights into the annotation process, (e.g., cognitive processes involved in annotation and their order during annotation), or leveraged in training a classifier using gaze patterns (e.g., fixations and saccades) as additional signals to improve subjectivity extraction and sentiment prediction.”

“Taken together, despite the central role that human annotation plays in training and evaluating sentiment analysis systems, relatively little work has directly examined the annotators themselves with regards to their instructions, experiences, and interpretation strategies.” [Prev invest]:

This is a powerful justification for your study and should be more strongly emphasized.

Consider elevating this sentence to begin the paragraph, and expanding briefly on why this oversight matters.

Thank you for pointing this out! We have since emphasised this in the beginning of the section:

“Despite the essential role annotators play in shaping datasets, relatively little attention has been paid to how they engage with annotation tasks in practice. Understanding annotator behaviour and experience is crucial for improving annotation quality and, by extension, the reliability of systems trained on such data.”

We also added to the last paragraph of this section:

“Taken together, despite the central role that human annotation plays in training and evaluating sentiment analysis systems, relatively little work has directly examined the annotators themselves with regards to their instructions, experiences, and interpretation strategies. This lack of attention limits our understanding of how annotation quality is shaped in practice, which impacts the reliability of sentiment models and the validity of their real-world application. Without insights into how annotators interpret and carry out the task, key assumptions about label consistency, task difficulty, and annotator intent may go unexamined – ultimately weakening the foundations of sentiment analysis systems.”

“Although human-annotated data is essential to sentiment analysis, the choices made during the annotation process are rarely examined in detail.” [Rationale]:

The opening sentence rightly underscores the importance of human-annotated data in sentiment analysis, but it could benefit from a more specific articulation of the problem.

---

## [Decision Letter · Decision Letter 1]

10 Sep 2025

Dear Dr. Hinds, 

Thank you for submitting your manuscript to PLOS ONE. After careful consideration, we feel that it has merit but does not fully meet PLOS ONE’s publication criteria as it currently stands. Therefore, we invite you to submit a revised version of the manuscript that addresses the points raised during the review process.

We look forward to receiving your revised manuscript.

Kind regards,

Wei Lun Wong

Academic Editor

PLOS ONE

Journal Requirements:

Reviewers' comments:

Reviewer's Responses to Questions

**Comments to the Author**

Reviewer #1: All comments have been addressed

Reviewer #2: All comments have been addressed

2. Is the manuscript technically sound, and do the data support the conclusions?

Reviewer #1: Yes

Reviewer #2: Yes

3. Has the statistical analysis been performed appropriately and rigorously?

Reviewer #1: Yes

Reviewer #2: Yes

4. Have the authors made all data underlying the findings in their manuscript fully available?

Reviewer #1: Yes

Reviewer #2: Yes

5. Is the manuscript presented in an intelligible fashion and written in standard English?

Reviewer #1: Yes

Reviewer #2: Yes

Reviewer #1: Thank you for the opportunity to review this rigorously conducted and highly relevant manuscript. The study addresses a critical, yet often overlooked, aspect of NLP research: the human annotator. The mixed-methods design is a particular strength, providing deep, explanatory insights that purely quantitative work often misses. The manuscript is well-structured, the methodology is sound, and the conclusions are well-supported by the data. It is a valuable contribution to the field.

Below are my comments, which are largely minor suggestions to further strengthen an already excellent manuscript:

1. Data Integrity & Transparency:

Exclusion Process: Please state the total number of participants initially recruited before exclusions (e.g., "Out of XXX initially recruited, 18 were excluded..."). This provides full transparency on attrition.

'Correct Response' Benchmark: Briefly clarify in the Methods that the "correct response" was strictly defined by alignment with the preliminary ground truth established by the three native speakers. A short sentence on the rationale for using this benchmark would be helpful.

2. Statistical Rigor:

Multiple Comparisons: The exploratory comparisons of inter-rater reliability across sentence types are informative. Please explicitly state in the figure caption or results text that these comparisons are exploratory and were not adjusted for multiple testing, acknowledging this as a limitation for this specific analysis.

3. Qualitative Depth:

Inter-coder Reliability: The thematic analysis is insightful. To further bolster its rigor, please report inter-coder reliability metrics (e.g., Cohen's Kappa on a sample) if a second coder was involved. If the analysis was conducted by a single coder, please acknowledge this as a standard limitation in the Methods.

Theoretical Discussion: The link to dual-process theory in the discussion is excellent. Please deepen this slightly by more explicitly mapping the identified annotation approaches (e.g., 'word-focused' -> Type 1 processing, 'empathic role-taking' -> Type 2 processing).

4. Language and Copyediting:

The manuscript is very well-written. Please perform a final proofread to correct minor formatting artifacts from the track changes process and ensure consistency. Specific items to check:

Page 24, Abstract: Correct "text-classificationnatural" to "natural" and "thntused" to "used".

Page 33, Procedure: Correct subject-verb agreement: "The succession... are" should be "The succession... is".

Page 122, Introduction: Correct the citation error [1, 2, 3],[4, 5, 6, ... ,33] to the intended format (likely [1-3]).

Consistency: Ensure consistent formatting of "Fig" vs. "Fig." and "Session" vs. "session" throughout.

5. Minor Clarifications:

Preliminary Ground Truth: In the Procedure section, briefly note that the native speakers validating the stimuli were independent raters (if they were).

Krippendorff's Alpha Interpretation: A brief interpretation of the alpha values (e.g., α=0.50-0.59 indicates "moderate" disagreement) in the context of sentiment annotation would aid readers.

This is a strong paper. My comments are intended as constructive suggestions to help you achieve the highest possible level of clarity and rigor in the final version. I congratulate the authors on a fine piece of work.

Reviewer #2: (No Response)

**Do you want your identity to be public for this peer review?** For information about this choice, including consent withdrawal, please see our Privacy Policy

Reviewer #1: No

Reviewer #2: No

---

## [Author Response · Author response to Decision Letter 2]

24 Sep 2025

Point by Point Response

Reviewer #1 Comments

Thank you for the opportunity to review this rigorously conducted and highly relevant manuscript. The study addresses a critical, yet often overlooked, aspect of NLP research: the human annotator. The mixed-methods design is a particular strength, providing deep, explanatory insights that purely quantitative work often misses. The manuscript is well-structured, the methodology is sound, and the conclusions are well-supported by the data. It is a valuable contribution to the field.

We greatly appreciate the positive comments, and we are glad you enjoyed the manuscript. We especially value your remarks on its rigor and relevance to the field. Thank you for taking the time to review our paper.

Below are my comments, which are largely minor suggestions to further strengthen an already excellent manuscript:

1. Data Integrity & Transparency:

Exclusion Process: Please state the total number of participants initially recruited before exclusions (e.g., "Out of XXX initially recruited, 18 were excluded..."). This provides full transparency on attrition.

Thank you for this comment! We agree full transparency on attrition is helpful, and we have checked this in more detail and expanded on the description, which now includes all participants who started the study (and decided not to participate), as well as different reasons for exclusion of successfully recruited participants. The relevant section in the manuscript is now updated as follows:

“With this requirement, we continued data collection until we had 53 participants per group, counting the first 53 who successfully completed both sessions. Out of 145 participants who started the study, 17 abandoned it before giving initial consent (i.e., decided not to participate), and were therefore not compensated. Out of the remaining 128 individuals successfully recruited, 22 were excluded for the following reasons: nine participants due to failing attention checks, 10 participants due to not completing session 2, and three participants due to over-recruitment. While exclusion due to failed attention checks or incomplete data was essential for maintaining data integrity, we recognize the ethical responsibility to ensure fair treatment of all participants. Accordingly, excluded participants were compensated pro rata for their time spent on the task, ensuring no participant was disadvantaged financially. Over-recruited participants were compensated on the same basis as included participants (see section Data collection and compensation).”

'Correct Response' Benchmark: Briefly clarify in the Methods that the "correct response" was strictly defined by alignment with the preliminary ground truth established by the three native speakers. A short sentence on the rationale for using this benchmark would be helpful.

Thank you – we have clarified this further in the Procedure section:

“The sentiment label of each sentence corresponds to the sentiment it was judged to express appropriately (by at least three independent native English speakers), and we refer to these labels as the initial ground truth. These ground truth labels were used as an indicator of instruction comprehension and adherence in the annotation task described below.”

This is also described briefly in the ‘Task duration and correct responses’ section, which we have now clarified by adding ‘from at least three native English speakers’:

“The time spent on the tasks and the proportion of correct responses (against the initial ground truth from at least three native English speakers) can be used as indications of participant engagement.” […] “Moreover, each participant scored above 40% correct, exceeding the 25% chance level for a four-option task, indicating reasonable engagement and comprehension of the instructions. However, the mean difference in the correct responses between the instructions groups is small, which may suggest low task engagement by participants in the Detailed group, or that the Minimal group spontaneously annotated the sentences in line with the detailed instructions.”

We also touch this topic in the ‘Instructions’ section in ‘Rationale for the current study’, by explaining that:

“While we do not suggest that detailed instructions eliminate all ambiguity or subjectivity in sentiment annotation, they are expected to reduce interpretative freedom and inconsistency regarding certain aspects of the annotation task, such as who's sentiment to prioritise or how to approach conflicting sentiments.”

We hope that highlighting these sections makes it clearer that the initial ground truth labels were used as indication of task engagement, particularly instruction adherence, and that following the detailed instructions should significantly reduce ambiguity in the sentences, thus making the initial ground truth labels a justifiable benchmark for this purpose.

2. Statistical Rigor:

Multiple Comparisons: The exploratory comparisons of inter-rater reliability across sentence types are informative. Please explicitly state in the figure caption or results text that these comparisons are exploratory and were not adjusted for multiple testing, acknowledging this as a limitation for this specific analysis.

Thank you for this observation! We agree this should be mentioned had we conducted statistical hypothesis testing for the different sentence types. However, this was not the case, as the comparisons presented here are merely descriptive, and as you mentioned, exploratory. This is due to the small number of sentences per sentence type, making formal hypothesis testing inappropriate at this level. We have clarified this in the section 'Inter-rater reliabilities in session 1' as follows:

"To answer RQ2, the inter-rater reliabilities were inspected descriptively for different sentence types (no formal statistical comparisons were conducted due to small number of sentences per sentence type)." [...] "The only clear numerical differences between the instructions groups were seen in agreement levels for Appraisal and especially for Two opinion holders and Two targets sentence types."

3. Qualitative Depth:

Inter-coder Reliability: The thematic analysis is insightful. To further bolster its rigor, please report inter-coder reliability metrics (e.g., Cohen's Kappa on a sample) if a second coder was involved. If the analysis was conducted by a single coder, please acknowledge this as a standard limitation in the Methods.

Thank you for this suggestion! We discussed this in our response to the first round of revisions with the other reviewer, explaining our position as follows:

We have followed Braun & Clark’s (2007, 2017) steps for reflexive thematic analysis, and due to the nature of this approach, the key component is for themes to be generated as a part of an active interpretation of the data, by the researcher. By nature, this kind of interpretivist, qualitative approach is conducted by one individual, unlike other approaches, such as a content analysis or framework analysis. Hence, there was not a reliability check nor any coding framework development, as this is not part of this methodology and would be invalid, as these align with more positivist and frequency-based approaches. This is exemplified, for instance, by Braun and Clarke’s (2023) distinction between ‘Big Q’ (their approach), and ‘small q’ approaches:

“Methodological incoherence beckons when researchers seemingly unknowingly mash together different approaches to TA (see Braun & Clarke, 2022a). There were several examples in the papers we reviewed of researchers using Big Q reflexive TA approach and procedures, with conceptually incoherent additions, such as small q consensus coding and measuring intercoder agreement, or adding a codebook development/recoding the data phase, or referencing both positivist notions of researcher bias and Big Q notions of reflexivity, or expressing concern for the accuracy and objectivity of the coding or the potential for misinterpreting the data (implying that correct interpretation is possible).”

References:

Braun V, Clarke V. Toward good practice in thematic analysis: Avoiding common problems and be(com)ing a knowing researcher. International Journal of Transgender Health. 2023;24: 1–6. doi:10.1080/26895269.2022.2129597

Theoretical Discussion: The link to dual-process theory in the discussion is excellent. Please deepen this slightly by more explicitly mapping the identified annotation approaches (e.g., 'word-focused' -> Type 1 processing, 'empathic role-taking' -> Type 2 processing).

Thank you for this comment! In the section ‘Discussion of thematic analysis findings’, we have added the following phrase to prime the reader for the more detailed discussion about how the identified annotation approaches map onto the processing types of the dual-process theory. We hope this makes it clearer:

"With these general characteristics, we discuss how the identified annotation approaches map onto the processing types of dual-process theories. The word-focused approach appears to involve mostly Type 1 processing..."

4. Language and Copyediting:

The manuscript is very well-written. Please perform a final proofread to correct minor formatting artifacts from the track changes process and ensure consistency. Specific items to check:

Page 24, Abstract: Correct "text-classificationnatural" to "natural" and "thntused" to "used".

We are grateful for your high attention to detail! We carefully checked both the PDF attached with these revisions and the version of the manuscript we submitted after the first round of revisions, but we are unable to locate these typos in our copies of the document. We will of course proofread the final version thoroughly. It is possible that what you observed may stem from a display or rendering issue (e.g., due to PDF encoding, font embedding, or text extraction from highlights/comments?) rather than errors in the underlying manuscript.

Page 33, Procedure: Correct subject-verb agreement: "The succession... are" should be "The succession... is".

Thank you – this is corrected now.

Page 122, Introduction: Correct the citation error [1, 2, 3],[4, 5, 6, ... ,33] to the intended format (likely [1-3]).

Thank you – this is another error we cannot locate in our versions of the documents. Hopefully if these are indeed part of the manuscript, we will find them upon final proof-reading.

Consistency: Ensure consistent formatting of "Fig" vs. "Fig." and "Session" vs. "session" throughout.

Thank you for your thoroughness! Again, we could not locate inconsistencies in Figures (all instances in our version of the document were ‘Fig’, while none were ‘Fig.’). However, we did find inconsistencies with ‘Session’ vs ‘session’, which are now corrected.

5. Minor Clarifications:

Preliminary Ground Truth: In the Procedure section, briefly note that the native speakers validating the stimuli were independent raters (if they were).

Thank you, this has been added to the description as follows:

“To ensure that these crowdworker-generated sentences fit the set criteria (e.g., positive sentiment expression via sarcasm, expression of a negative sentiment via a rhetorical question), they were independently evaluated for validity by three native English speakers and included in the final dataset based on majority vote.”

Krippendorff's Alpha Interpretation: A brief interpretation of the alpha values (e.g., α=0.50-0.59 indicates "moderate" disagreement) in the context of sentiment annotation would aid readers.

Thank you - this is indeed helpful for the reader. We have contextualised our findings more in the section 'Inter-rater reliabilities in session 1', after presenting the first results regarding inter-rater reliability:

"It is worth pointing out that commonly cited thresholds for Krippendorff’s alpha are values between 0.667 and 0.800 for tentative conclusions, whereas over 0.800 are generally considered reliable [74] (p. 241-242). Our findings reveal low inter-annotator reliability overall, which likely reflects inherent ambiguity in sentiment judgments, but may also be indicative of inconsistencies in instruction adherence. However, the actual reliability thresholds should be considered based on the intended purpose, as pointed out by Krippendorff [74] and others [75,76]. Furthermore, inter-rater reliabilities tend to decrease with increasing rater sample sizes [28,77] as more diverse viewpoints introduce greater potential for disagreement. Thus, our reliability estimates are not simply indicative of low annotation agreement, but also a natural consequence of the large number of annotators involved in our study. This is in contrast to studies with only two or a handful of annotators per item, which may artificially inflate reliability estimates by underrepresenting the true range of interpretations."

This is a strong paper. My comments are intended as constructive suggestions to help you achieve the highest possible level of clarity and rigor in the final version. I congratulate the authors on a fine piece of work.

We are very grateful for your time, helpful comments and thoroughness! Your kind suggestions have certainly helped us improve the manuscript.

Reviewer #1 Comments from the manuscript (the PDF)

“Disambiguating sentiment annotation: A mixed methods investigation of annotator experience and impact of instructions on annotator agreement” [Title]

The paper has been significantly improved and the authors have responded carefully and respectfully to all of the reviewer's comments. Changes made include:

Improved structure and readability

Added methodological detail

Stronger theoretical definitions

More complete reporting of statistical results

More transparency in the ethics and methods section

Thank you for the summary of the improvements!

“The time spent on the tasks and the proportion of correct responses (against the initial ground truth) can be used as indications of participant engagement.” [Results: Task duration and correct responses]

Please clarify the specific criteria used to define a 'correct response'. Was it strictly based on the preliminary ground truth label derived from the three native speakers? Given that the study explores ambiguity, briefly discuss the implications of using this benchmark, especially for sentences where multiple interpretations are valid

Responded to above in the ‘Reviewer #1 Comments (in our response beginning with “Thank you – we have clarified this further in the Procedure section:

“The sentiment label of each sentence corresponds to the sentiment it was judged to express appropriately”)

“Inter-rater reliabilites, quantified as Krippendorff’s alpha for the full dataset of 252 sentences were 0.50 in the Detailed group and 0.59 in the Minimal group. “[Results: Inter-rater reliabilities in Session 1]

Consider adding a brief interpretation of these alpha values in the context of sentiment annotation.

For example, does α = 0.50 indicate 'moderate' disagreement due to inherent ambiguity, or does it reflect a problem? Reference to common interpretation scales (e.g., Krippendorff, 2004) would be helpful.

Thank you for noting this, we have responded to above in the ‘Reviewer #1 Comments (in our response beginning with “Thank you - this is indeed helpful for the reader. We have contextualised our findings more in the section 'Inter-rater reliabilities in session 1',…”).

“I think there was an example in the instructions where "England beat France at the football" was supposed to be positive, but "England lost to France" was negative. I really don't care about football, England, or France, and these are factual statements not opinions. In the task there were "Norway beat Sweden at the ice hockey" and "Norway lost to Sweden" so I marked these as neutral. (P30)” [Results: Thematic analysis of annotation experiences: Detailed group]

This is a powerful quote that highlights a key challenge: annotators may disregard instructions due to personal beliefs (e.g., 'factual statements not opinions'). Consider adding a sentence in the discussion to emphasize that instructions must not only be detailed but also effectively motivate annotators to adhere to them, especially when they conflict with personal intuition

Than

---

## [Decision Letter · Decision Letter 2]

22 Oct 2025

Disambiguating sentiment annotation: A mixed methods investigation of annotator experience and impact of instructions on annotator agreement

PONE-D-25-28779R2

Dear Dr. Hinds, 

We’re pleased to inform you that your manuscript has been judged scientifically suitable for publication and will be formally accepted for publication once it meets all outstanding technical requirements.

Kind regards,

Wei Lun Wong

Academic Editor

PLOS ONE

Additional Editor Comments (optional):

Reviewers' comments:

Reviewer's Responses to Questions

**Comments to the Author**

Reviewer #2: All comments have been addressed

Reviewer #3: All comments have been addressed

2. Is the manuscript technically sound, and do the data support the conclusions?

Reviewer #2: Yes

Reviewer #3: Yes

3. Has the statistical analysis been performed appropriately and rigorously?

Reviewer #2: Yes

Reviewer #3: Yes

4. Have the authors made all data underlying the findings in their manuscript fully available?

Reviewer #2: Yes

Reviewer #3: Yes

5. Is the manuscript presented in an intelligible fashion and written in standard English?

Reviewer #2: Yes

Reviewer #3: Yes

Reviewer #2: (No Response)

Reviewer #3: The authors have diligently revised the manuscript in response to the previous round of reviews. It's clear that careful attention was paid to Reviewer #1's comments, and the manuscript is significantly strengthened as a result. The additions regarding data integrity and transparency are welcome. Specifically, the clarification of the participant exclusion process, including the initial recruitment numbers and reasons for exclusion, provides valuable context . Similarly, defining the "correct response" benchmark based on the initial ground truth established by independent native English speakers enhances methodological clarity.

In terms of statistical rigor, the authors have appropriately clarified that the comparisons of inter-rater reliability across sentence types were descriptive and exploratory, thereby addressing the concern about multiple comparisons adjustments in this specific context . The added interpretation of the obtained Krippendorff's alpha values, contextualizing them within common benchmarks and considering the impact of the large annotator sample size, is also a helpful addition for readers .

The qualitative depth has been enhanced, particularly through the more explicit mapping of the identified annotation approaches (word-focused, holistic, empathic) onto Type 1 and Type 2 processing within the dual-process theory framework . This deepens the theoretical contribution regarding annotator behavior.

The requested copyediting fixes and minor clarifications appear to have been successfully implemented, including corrections to grammar , consistent terminology (e.g., "Session") , noting the independence of raters validating stimuli , defining "soft labels" upon introduction , and incorporating the point about motivating instruction adherence even when it conflicts with intuition . The discussion regarding the generalizability limitation due to the use of crowdworkers has also been added appropriately .

The study remains a strong contribution, offering valuable insights into the complexities of sentiment annotation through its robust mixed-methods design and the creation of the useful AmbiSent dataset. The investigation into how instructions impact agreement and the exploration of annotator strategies are highly relevant to the NLP community.

For future work, it would be valuable to explore how task engagement and cognitive framing could be enhanced through adaptive instruction systems or dynamic feedback loops that sustain annotator attention and align task framing with underlying cognitive tendencies. Extending this research beyond crowdworkers and native English speakers to include multilingual, cross-cultural, or domain-expert annotators could also illuminate how cultural context and professional expertise modulate interpretive ambiguity. Finally, leveraging the AmbiSent dataset to train or fine-tune machine learning models on soft labels could empirically demonstrate how modeling uncertainty improves the robustness and fairness of sentiment classifiers. If the authors wish, they can include these points in the future work section.

**Do you want your identity to be public for this peer review?** For information about this choice, including consent withdrawal, please see our Privacy Policy

Reviewer #2: No

Reviewer #3: No

---

## [Editor Report · Acceptance letter]

PONE-D-25-28779R2

PLOS ONE

Dear Dr. Hinds,

I'm pleased to inform you that your manuscript has been deemed suitable for publication in PLOS ONE. Congratulations! Your manuscript is now being handed over to our production team.

Kind regards,

on behalf of

Dr. Wei Lun Wong

Academic Editor

PLOS ONE